# PROPERTIES FROM MECHANISMS:
# AN EQUIVARIANCE PERSPECTIVE ON IDENTIFIABLE REPRESENTATION LEARNING

**Kartik Ahuja**[*]**, Jason Hartford**[*]**& Yoshua Bengio**
Mila - Quebec AI Institute, Université de Montréal
Quebec, Canada
{kartik.ahuja,jason.hartford,yoshua.bengio}@mila.quebec

## ABSTRACT

A key goal of unsupervised representation learning is "inverting" a data generating process to recover its latent properties. Existing work that provably achieves this goal relies on strong assumptions on relationships between the latent variables (e.g., independence conditional on auxiliary information). In this paper, we take a very different perspective on the problem and ask, "Can we instead identify latent properties by leveraging knowledge of the mechanisms that govern their evolution?" We provide a complete characterization of the sources of non-identifiability as we vary knowledge about a set of possible mechanisms. In particular, we prove that if we know the exact mechanisms under which the latent properties evolve, then identification can be achieved up to any equivariances that are shared by the underlying mechanisms. We generalize this characterization to settings where we only know some hypothesis class over possible mechanisms, as well as settings where the mechanisms are stochastic. We demonstrate the power of this mechanism-based perspective by showing that we can leverage our results to generalize existing identifiable representation learning results.These results suggest that by exploiting inductive biases on mechanisms, it is possible to design a range of new identifiable representation learning approaches.

## 1 INTRODUCTION

Modern unsupervised learning techniques can generate images of our world with intricate detail (e.g. Karras et al., 2019; Song et al., 2020; Razavi et al., 2019), and yet, the latent representations from which these images are generated remain entangled and challenging to interpret (Schölkopf et al., 2021; Locatello et al., 2019). At the same time, the success of pre-trained transformers (Radford et al., 2021; Brown et al., 2020) shows that advances in our ability to extract latent representations can lead to dramatic improvements in the sample complexity of downstream tasks (Bengio & LeCun, 2007; Bengio et al., 2013). In order to consistently replicate this success, we need methods that can reliably invert the data generating process into its underlying generative factors. This is a challenging task because unsupervised representation learning with independent and identically distributed data is hopelessly *unidentified*: even in the nice case in which observations, $x$, are generated with independent latent variables, $p(z) = \prod_i p(z_i)$, there exists an infinitely many distributions with independent latents $\tilde{p}(z)$ that are consistent with the observed marginal distribution, $p(x)$, and only one of them corresponds to the true latent distribution $p(z)$ (Locatello et al., 2019; Khemakhem et al., 2020a).

If we want to build systems which are able to provably *identify* [1] the true generative factors $z$ that generated our observed data $x = g(z)$ for some observation model $g$, then we need to exploit structural assumptions that constrain this set of possible distributions. Most of the prior work that can provide such guarantees was developed in the independent component analysis (ICA) literature. The key ICA assumption is that the latent factors are (conditionally) independent and non-Gaussian. Then, if the observation model, $g : \mathcal{Z} \to \mathcal{X}$ is linear and invertible, $z = g^{-1}(x)$ is identified (Comon,

---

[*]equal contribution, author order selected randomly.
[1]A problem is identified if the true generative factors are a unique solution in the infinite data limit.

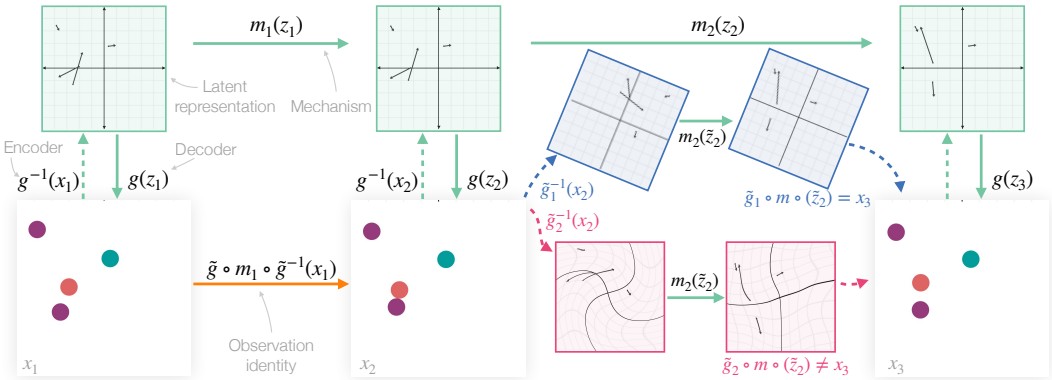

Figure 1: This simple data generating process illustrates that if we know the set of mechanisms that govern the evolution of an environment, this constrains the set of possible representations to any equivariances of these mechanisms. At each time step, we observe an environment of bouncing balls in pixel space as images, $x_t$. These images are produced by some rendering engine, $g$, as a function of the true latent representation $z_t$, which in this case gives positions and velocities of each ball (illustrated by the location and length of the arrows in the latent representation). At each time step, the state evolves according to a mechanism, $m_t$. Any candidate model that is consistent with the observed data has to satisfy the observation identity. If an encoder produces either the true representation (shown in green), or a representation transformed by some equivariance of the mechanism (e.g. the blue rotated representation) then the observation identity is satisfied. However, models that produce representations that are arbitrary transformations of the true representation (e.g. the pink warped representation) can be discarded as they are not consistent with the observation identity.

1994), and for nonlinear $g$ there are a number of recent approaches that leverage non-stationarity in the distributions over $z$ to identify $g^{-1}$ (Hyvarinen & Morioka, 2016; 2017; Hyvarinen et al., 2019; Khemakhem et al., 2020a). These results give a tantalizing demonstration that representation learning with identification guarantees is possible, but the requirement that the latent factors are statistically (conditionally) independent is limiting[2] (Higgins et al., 2018; Schölkopf et al., 2021).

In this paper we take a very different approach. We study how the mechanisms that govern an environment's evolution constrain the set of possible latent representations that are consistent with the data. As a simple concrete example, consider the bouncing ball environment shown in Figure 1. The latent state can be completely described by a vector, $z$, containing the position, velocity and acceleration of the balls[3], and given this latent state, the images, $x$, shown in Figure 1, are produced via some rendering engine $g : \mathcal{Z} \to \mathcal{X}$. Our task is to leverage sequences of observations $x_1, \ldots, x_T$ and knowledge of $m$ to recover $z_1, \ldots, z_T$. If we can show that this task has a unique solution that is consistent with the observations and mechanisms, then the problem is identified.

Our main result is that when we know the true mechanism, $m$, the system is identified up to any equivariances of the mechanism; or equivalently, in Section 2.2 we prove that we can identify $z$ up to any invertible transformation $a : \mathcal{Z} \to \mathcal{Z}$ that commutes with $m$, such that the composition $m \circ a = a \circ m$. For example, in the bouncing balls environment shown in Figure 1, the laws of physics are equivariant with respect to your choice of units of measurement—changing from representing $z$ in meters to inches leaves the output of the mechanism unchanged up to a corresponding unit change— and hence we can only hope to identify $z$ up to some scaling function $a(z)$ which corresponds to an arbitrary choice of units of measurement. Interestingly, when the environment evolves according to multiple known mechanisms, the sources of non-identifiability are even further constrained: such a system is identified up to equivariances that are shared by all $n$ mechanisms.

---

[2]As a simple example of dependence between latent variables, assume the bouncing balls shown in Figure 1 have different masses indicated by their colors. If the initial conditions were such that all the balls have the same momentum, then mass and velocity will be inversely correlated.

[3]A complete generative model of these images would also need to track the colors and shapes of the elements; we will return to this issue in the discussion of the results.

Perfectly knowing the true mechanisms and when they are applied is unlikely, but in Section 2.3 we show that we can relax that assumption to a setting where we instead know a hypothesis class of possible mechanisms that could have been applied. This weaker assumption leads to an additional source of non-identifiability: the mechanisms in our hypothesis class can *imitate* each other if there exists an invertible transformation $a$ such that for any two mechanisms $m_1$ and $m_2$ in our hypothesis class, $m_2 = a^{-1} \circ m_1 \circ a$. For example, if we are in a setting where our hypothesis class includes both a product mechanism, $m_1(z) = \prod_i z_i$, and a sum mechanism, $m_2(z) = \sum_i z_i$, then $m_1$ can imitate $m_2$ if $a(z) = \exp(z)$, since $\sum_i z_i = \log(\prod_i \exp(z_i))$. As before, this result is complete, in the sense that these two sources of non-identifiability—equivariance and imitation—are the only sources of non-identifiability in such systems. This gives us a natural way of thinking about the way knowledge of deterministic mechanisms constrains a representation learning task: with complete knowledge, the only source of non-identifiability is any equivariances inherent in the mechanism, but by allowing a hypothesis class of possible mechanisms, we introduce potential non-identifiability via imitation. That said, the relationship between the size of the hypothesis class and the size of the set of $a$'s that commute via imitation is not necessarily monotonic: for environments governed by multiple mechanisms, a larger hypothesis class can lead to fewer imitations.

Section 3, shows that we can derive analogous results for stochastic mechanisms, $m(z, U)$, where the mechanism defines a conditional distribution $p(z_{t+1}|z_t)$. This generalization gives us a way of comparing our mechanism-based perspective with existing identifiability results. We demonstrate this in Section 4 by showing that it is possible to view the distributional assumptions made in Klindt et al. (2020) as a particular choice of mechanism, and by doing so, we can leverage our theory to give alternative proofs of these results. This strategy required weaker distributional assumptions, thereby generalizing their result. Finally, we give a mechanism-based perspective on the related work in Section 5 and Section 6 concludes with a discussion of the open problems that need to be addressed in order to reliably leverage this approach in practice.

## 2 MECHANISM BASED IDENTIFICATION

### 2.1 DATA GENERATION PROCESS

The state of the system at time $t \in \{1, \cdots, T\}$ is given by $z_t \in \mathcal{Z} \subseteq \mathbb{R}^d$. At each time $t$, we observe $g(z_t) = x_t \in \mathcal{X} \subseteq \mathbb{R}^n$, which is some transformation of the latent $z_t$. We can think of $g : \mathcal{Z} \to \mathcal{X}$ as a function that transforms the (typically low dimensional) state variables to the (typically high dimensional) observed variables; for example, in the bouncing ball environment described in the introduction, $g$ is the rendering engine that produces the images shown in Figure 1. We assume $g$ is injective [4] with respect to $\mathbb{R}^n$—i.e. $g(z_1) = g(z_2)$ implies $z_1 = z_2$; or equivalently, any change to the underlying state, $z$, is reflected in some pixel-level change to the observation $x$—and we make $g$ bijective by restricting its inverse $g^{-1}$ to any $x$ on the data manifold which we denote $\mathcal{X}$ (i.e. $\mathcal{X}$ is the image of $g$). The state transition from time $t$ to $t+1$ is governed by a mechanism $m_t : \mathcal{Z} \to \mathcal{Z}$. There may be multiple mechanisms in a given environment. For example in Figure 1 the transition from $z_1$ to $z_2$ does not involve any collisions so the state evolves according to Newton's first law of motion (Newton, 1687); the transition from $z_2$ to $z_3$ involves a collision between two of the balls that is described by Newton's third law. Together $m_t$ and $g$ describe the data-generation from time $t$ to $t+1$ as follows,

$$x_t \leftarrow g(z_t), \qquad z_{t+1} \leftarrow m_t(z_t). \tag{1}$$

If we fix the initial conditions, $z_0$, this data generation process is deterministic. In Section 2.2, we provide results when the underlying mechanism is known and in Section 2.3, we we extend those results to the case when the mechanisms are not known. In Section 3, we extend our results to stochastic mechanisms, $m(Z_t, U)$ that take samples from some distribution $U \sim \mathrm{Uniform}(0, 1)$ as input. These stochastic mechanisms can represent any conditional distribution $P(Z_{t+1}|Z_t)$ (Austin (2015, Lemma 3.1))

### 2.2 IDENTIFYING ENCODERS WHEN THE UNDERLYING MECHANISM IS KNOWN

We begin in the simplest version of the system described by equation (1): assume that at each time $t$ the same mechanism $m : \mathcal{Z} \to \mathcal{Z}$ is used, and we know this mechanism. From these assumptions we

---

[4]for ball collision example, if Z equals position and velocity, we can achieve injectivity by frame stacking.

can derive an identity that describes how the observations $x_t$ and $x_{t+1}$ are related,

$$z_{t+1} = m(z_t), \quad g^{-1}(x_{t+1}) = m \circ g^{-1}(x_t), \quad x_{t+1} = g \circ m \circ g^{-1}(x_t). \qquad (2)$$

It may be helpful to think of this identity as describing an autoencoder where we require that the encoder $g^{-1}(x_t)$ inverts the data generating process to produce some latent $z_t$ from $x_t$, and that the decoder has to reproduce $x_{t+1}$ from $m(z_t)$; i.e. the representation transformed by the mechanism. Importantly, this identity describes the true data generating process, rather than some model of it. Our hypothesis class over the possible encoder / decoder functions is the set, $\mathcal{G}$, of all bijective functions from $\mathcal{Z} \to \mathcal{X}$. By assuming that bijectivity we are essentially assuming that the reconstuction task is solved:[5] $\mathcal{G}$ is the set of all autoencoders that perfectly reproduce the data, such that for any $x$ on the data manifold $\mathcal{X}$, and any encoder / decoder pair $(\tilde{g}^{-1}, \tilde{g})$ with $\tilde{g} \in \mathcal{G}$, their composition is the identity function, $x = \tilde{g} \circ \tilde{g}^{-1} \circ x$. Because we assumed that the true $g$ is bijective, we know that it is in our search space, $\mathcal{G}$. We can constrain this set using our knowledge of the mechanism by only considering solutions that also satisfy the identity given in equation 2, such that for every pair of observations $(x_t, x_{t+1})$, an analogous identity holds,

$$x_{t+1} = \tilde{g} \circ m \circ \tilde{g}^{-1}(x_t) \qquad (3)$$

Now suppose that we have access to observations $x_t$ from the entire set $\mathcal{X}$, then the above identities have to hold for all $x_t, \in \mathcal{X}$ with corresponding $x_{t+1}$ from equation 2, so we can conclude that the following functions are equal,

$$g \circ m \circ g^{-1} = \tilde{g} \circ m \circ \tilde{g}^{-1} \qquad (4)$$

This relationship will hold for any of the possible decoders $\tilde{g}$ (and corresponding encoders $\tilde{g}^{-1}$) that are observationally equivalent given our assumptions. We denote the set of all such decoders, $\mathcal{G}_{\mathsf{id}} = \{\tilde{g} \mid \tilde{g} \in \mathcal{G}, g \circ m \circ g^{-1} = \tilde{g} \circ m \circ \tilde{g}^{-1}\}$. If $\mathcal{G}_{\mathsf{id}} = \{g\}$ then we have shown that the problem is exactly identified, which means that if we manage to find a $\tilde{g}$ that satisfies equation 2, then $\tilde{g} = g$; but if $\mathcal{G}_{\mathsf{id}}$, also includes other functions $\tilde{g} \neq g$, then these functions are the sources of non-identifiability.

**Equivariant mechanisms** To see an example of a setting where the problem is not exactly identified, consider a mechanism which is *equivariant* with respect to some bijective transformation $a : \mathcal{Z} \to \mathcal{Z}$. A mechanism $m$ is said to be *equivariant* w.r.t $a$ if $a \circ m = m \circ a$. For example, a mechanism derived from Newtonian mechanics may be equivariant with respect to your choice of units of measurement, such that $m(cz) = cm(z)$ for some scaling constant $c$. Similarly, if a mechanism transforms sets of items, any permutation of $z$ would lead to a corresponding permutation of the mechanism's output.

If we have a mechanism that is equivariant with respect to some transformation $a$ (where $a$ is not identity map), that implies that there exists a function $\tilde{g} \neq g$ in $\mathcal{G}_{\mathsf{id}}$, so the problem is not exactly identified. We can see this as follows,

$$g \circ m \circ g^{-1} = g \circ a^{-1} \circ a \circ m \circ g^{-1} = g \circ a^{-1} \circ m \circ a \circ g^{-1} = \tilde{g} \circ m \circ \tilde{g}^{-1}$$

where the first equality uses the fact that $a^{-1} \circ a$ is the identity function, the second applies the definition of equivariance, and the final equality defines $\tilde{g} := g \circ a^{-1}$ and $\tilde{g}^{-1} := a \circ g^{-1}$. This shows that if the mechanism is equivariant, an encoder, $\tilde{g}^{-1}$, can output a transformed $\tilde{z} = a(z)$, and the decoder, $\tilde{g}$, inverts this transformation before producing its output, thereby leaving the observed variables, $x$, unchanged. If we denote the set of all equivariances of the mechanism $\mathcal{E} = \{a \mid a \text{ is a bijection}, \ a \circ m = m \circ a\}$, then we can define the set of all such sources of non-identification as, $\mathcal{G}_{\mathsf{eq}} = \{\tilde{g} \mid \tilde{g} = g \circ a^{-1}, \ a \in \mathcal{E}\}$. This is a natural source of non-identification: given that we are relying on the mechanism to constrain the encoder $\tilde{g}^{-1}$, it is unsurprising we cannot prevent transformations that are not affected by the mechanism. The more interesting observation, which we will show below, is that this set is the *only* source of non-identification when the mechanism is known, and hence we recover the true $z$ up to equivariances in the mechanism.

To state this theorem, we first define the notion of identifiability up to an equivalence class defined by a family of bijections $\mathcal{A}$, where $a \in \mathcal{A}$ is a map $a : \mathcal{Z} \to \mathcal{Z}$.

**Definition 1.** *Identifiability up to $\mathcal{A}$. If the learned encoder $\tilde{g}^{-1}$ and the true encoder $g^{-1}$ are related by some bijection $a \in \mathcal{A}$, such that $\tilde{g}^{-1} = a \circ g^{-1}$ (or equivalently $\tilde{g} = g \circ a^{-1}$), then $\tilde{g}^{-1}$ is said to learn $g^{-1}$ up to bijections in $\mathcal{A}$. We denote this $\tilde{g}^{-1} \sim_{\mathcal{A}} g^{-1}$.*

---

[5]This is obviously a strong assumption—learning autoencoders that perfectly reconstructed the data is not at all easy—but it focuses the discussion on the identification issues that remain after reconstruction is solved.

Suppose, for example, $\mathcal{A}$ is a family of a permutations. Identifiability up to $\mathcal{A}$ implies that the true latent variables will be recovered, but that they would not necessarily be ordered in the same way as they were in the original data generation process. In this setting where the mechanism, $m$, is known, the following theorem shows that $\mathcal{A}$ is just $\mathcal{E}$, the set of equivariances of $m$.

**Theorem 1.** *If the data generation process follows equation 2, the encoders that solve the observation identity in equation 4 identify true encoder $g^{-1}$ up to equivariances of $m$ ($\tilde{g}^{-1} \sim_{\mathcal{E}} g^{-1}$).*

To prove the above theorem, we need to establish that the set of solutions to the identity in equation 4 and the set of maps derived from equivariances are equal $\mathcal{G}_{\text{id}} = \mathcal{G}_{\text{eq}}$. The proof is given in Section A.1 of the appendix. From Theorem 1, we can derive a number of observations. First, notice that if we have a standard autoencoder[6] then the mechanism is the identity map, $m(z) = z$, and its set of equivariances $\mathcal{E}$ is any invertible function $a$, and hence Theorem 1 shows that the encoder is essentially unconstrained. However, if the mechanism is any non-trivial function, $m(z') \neq z'$ for some $z'$, then the space of possible encoders is significantly reduced to just those invertible transformations that commute with $m$. If the system involves multiple known mechanisms, $\{m_1, \ldots, m_T\}$, where at each time $z_{t+1} = m_t(z_t)$, then the encoder is even further constrained. Define the set of all the mechanisms that are used at least once in the evolution of the system as $\mathcal{M}^* = \cup_{t=1}^T \{m_t\}$. Suppose $\mathcal{E}^i$ denotes the equivariances of $m_i \in \mathcal{M}^*$. Define the equivariances that are shared across all the mechanisms as $\mathcal{E}^* = \cap_i \mathcal{E}^i$;

**Corollary 1.** *If the data generation process follows equation 1, then the encoders that satisfy observation identity in equation 4 for all $m \in \mathcal{M}^*$ identify true encoder $g^{-1}$ up to the equivariances shared across all the mechanisms in $\mathcal{M}^*$, $\mathcal{E}^*$ ($\tilde{g}^{-1} \sim_{\mathcal{E}^*} g^{-1}$).*

The proof of the above claim is in Section A.3. This corollary implies a blessing that comes with more complex environments: if an object is transformed by multiple mechanisms which are diverse (in the sense that they share few equivariances), then it becomes easier to identify. Given access to inputs and outputs of a mechanism, if we cannot tell apart whether some transformation was applied to the input or the output, then the mechanism is equivariant with respect to the transformation. Together Theorem 1 and Corollary 1, state that we can learn to invert the data generation process but we cannot distinguish latents that were transformed by equivariances shared across the mechanisms.

### 2.2.1 IDENTIFYING ENCODERS FOR AFFINE MECHANISMS

Theorem 1 shows us that an encoder $\tilde{g}^{-1}$ is identified up to any equivariances of the known mechanism, but given some mechanism, it does not tell us what equivariances it may exhibit. This section gives an example of how one might go about finding all sources of equivariance for a given mechanism. We derive the equivariances for affine mechanisms, and in doing so we show conditions under which affine mechanisms lead to identification up to some fixed offset. Affine mechanisms are broadly applicable because with a sufficiently short time interval, they approximate a wide variety of physical systems as the Euler discretization of some linear ordinary differential equation. In such systems, the mechanism is given by $m(z) = Mz_t + b_t$ where $M \in \mathbb{R}^{d \times d}$ is an invertible diagonalizable matrix (with eigendecomposition given as $M = S\Lambda S^{-1}$, where $S$ is the matrix of eigenvectors and $\Lambda$ is a diagonal matrix of eigenvalues), $b_t \in \mathbb{R}^d$ is the offset parameter at time $t$, and the analog of equation 2 is,

$$x_{t+1} = g(Mg^{-1}(x_t) + b_t).$$

We search for some encoder $\tilde{g}^{-1}$ such that this relationship holds for all $x$ and $t$. Define an offset function $o(z) = z + p$, where the offset function shifts the latent by a vector $p$. Define $\mathcal{O}$ to be the set of all the offset functions. We show that the encoder is identified up to $\mathcal{O}$ when we have at least two distinct offset terms $b_t$ and a regularity condition (Assumption 2).[7]

**Assumption 1.** *In the data generation process in equation 1, we set $m(z_t) = Mz_t + b_t$, where $M$ is invertible and diagonalizable. We assume that the offset $b_t$ takes at least $d + 1$ distinct values, which we denote by $\{b^1, \cdots, b^{d+1}\}$. The set $\{b^2 - b^1, \cdots, b^{d+1} - b^1\}$ of vectors is linearly independent.*

**Assumption 2.** *$a : \mathcal{Z} \to \mathcal{Z}$ is analytic and satisfies the following assumption. For each component $i \in \{1, \cdots, d\}$ of $a_i(z)$ and each $b \in \mathbb{R}^d$, define the set $\mathcal{S}^{ij} = \{\theta \mid \nabla_j a_i(z + b) = \nabla_j a_i(z) + \nabla_j^2 a_i(\theta)b, z \in \mathbb{R}^d\}$. Each set $\mathcal{S}^{ij}$ has a non-zero Lebesgue measure in $\mathbb{R}^d$.*

---

[6]With the constraint that the encoder is the inverse of the decoder such that $\tilde{g}^{-1}$ is bijective.

[7]We conjecture that the regularity condition holds for all analytic functions and is thus not needed. Since we do not have a proof of this claim, we include it as an assumption.

**Theorem 2.** *If the data generation process follows equation 1 with affine mechanisms, $m(z) = Mz_t + b_t$, Assumptions 1, 2 hold, the eigenvalues of the mechanism $M$ are all distinct, and each component of $S^{-1}(b^i - b^j)$ is non-zero for some $i \neq j$, then the encoders that solve the observation identity in equation 4 identify true encoder $g^{-1}$ up to offsets $\mathcal{O}$ such that $\tilde{g}^{-1} \sim_{\mathcal{O}} g^{-1}$.*

The proof is given in Section A.4 in the Appendix. The above theorem shows the power of using multiple mechanisms. It can be shown that if there is only one mechanism, then we cannot do better than linear identification. However, if we use two mechanisms as is the case in the above theorem, the constraint of shared equivariances (Theorem 1 and Corollary 1) enforces almost exact identifiability (only offset-based errors remain).

## 2.3 IDENTIFYING ENCODERS WHEN THE MECHANISMS ARE NOT KNOWN

We have seen in the previous section that with complete knowledge of the mechanisms under which a system evolves, we can learn an encoder up to equivariances. In practice, however, we are unlikely to have such complete knowledge. In this section, the system still evolves according to some deterministic mechanism, $z_{t+1} \leftarrow m_t(z_t)$, but we assume that you only know some hypothesis class $\mathcal{M}$ of possible mechanisms which could have been used, without knowing which $m_t \in \mathcal{M}$ is used at every time step.

A candidate solution now needs to propose both an encoder $\tilde{g}^{-1}$ and a mechanism $\tilde{m}_t \in \mathcal{M}$ for every $(x_t, x_{t+1})$ pair such that,

$$x_{t+1} = \tilde{g} \circ \tilde{m}_t \circ \tilde{g}^{-1}(x_t). \tag{5}$$

As before, this relationship holds for all $x_t \in \mathcal{X}$, where $x_{t+1}$ is generated from $m_t$, so any candidate solution that is consistent with the $x$'s that we observe must satisfy

$$g \circ m_t \circ g^{-1} = \tilde{g} \circ \tilde{m}_t \circ \tilde{g}^{-1}. \tag{6}$$

We partition the hypothesis class of mechanisms, $\mathcal{M} = \mathcal{M}^* \cup \mathcal{M}'$, into the mechanisms that are used at least once in the evolution of the system, $\mathcal{M}^*$ ($\mathcal{M}^* = \cup_{t=1}^{T} \{m_t\}$), and mechanisms which are hypothesized but not used, $\mathcal{M}'$. We define the set of all decoders $\tilde{g}$ (with corresponding encoders $\tilde{g}^{-1}$) that solve equation 6 across all the time steps as as $\tilde{\mathcal{G}}_{\mathsf{id}} = \{\tilde{g} \mid \tilde{g} \text{ is a bijection }, \text{ for each } m_t \in \mathcal{M}^*, \exists \tilde{m}_t \in \mathcal{M}, \text{ such that } g \circ m_t \circ g^{-1} = \tilde{g} \circ \tilde{m}_t \circ \tilde{g}^{-1}\}$. This set looks very much like the set $\mathcal{G}_{\mathsf{id}}$ that we defined in Section 2.2, but the fact that we have to select $\tilde{m} \in \mathcal{M}$ rather than knowing the true $m$ implies a new source of non-identifiability: imitator mechanisms.

**Definition 2.** *Equivariances and imitators w.r.t $\mathcal{M}$. Define a set of functions $\tilde{\mathcal{E}}$ that satisfy commutativity w.r.t the set of mechanisms $\mathcal{M}$ in the following sense. The set $\tilde{\mathcal{E}}$ comprises of all the bijections, $a(\cdot)$, that satisfy the following condition. If for each $m_1 \in \mathcal{M}^*$, $\exists m_2 \in \mathcal{M}$ such that $a \circ m_1 = m_2 \circ a$, then $a \in \tilde{\mathcal{E}}$.*

The set $\tilde{\mathcal{E}}$ consists of two types of elements. We illustrate this through an example of set $\mathcal{M} = \mathcal{M}^* = \{m^1, m^2\}$. If $a$ is a bijection that commutes with both $m^1$ and $m^2$, i.e., $a \circ m^1 = m^1 \circ a$ and $a \circ m^2 = m^2 \circ a$, then $a \in \tilde{\mathcal{E}}$. From this we can see that $\tilde{\mathcal{E}}$ consists of elements in the intersection of the equivariances of the respective mechanisms. Alternatively, if $a$ satisfies $a \circ m_1 = m_2 \circ a$ and $a \circ m_2 = m_1 \circ a$, then we say $m_2$ "imitates" $m_1$ and vice-versa because you can produce $m_1$'s output from $m_2$ for any $z$ using the following relationship $m_1 = a^{-1} \circ m_2 \circ a$. Further simplifcation of this yields that $a^2 = a \circ a$ is an equivariance of both $m_1$ and $m_2$. This example shows that when we know the list of mechanisms $\mathcal{M} = \mathcal{M}^*$ but do not know which mechanism is used, the set $\tilde{\mathcal{E}}$ can be expressed in terms of the equivariances of the mechanisms. For further details see the Appendix Section A.11. Define the set of maps that are identified up to $\tilde{\mathcal{E}}$ as $\tilde{\mathcal{G}}_{\mathsf{eq}} = \{\tilde{g} \mid \tilde{g} = g \circ a^{-1}, a \in \tilde{\mathcal{E}}\}$.

**Theorem 3.** *If the data generation process follows equation 1, then the set of all the encoders that satisfy equation 6 identify true encoders $g^{-1}$ up to equivariances and imitators w.r.t $\mathcal{M}$, $\tilde{\mathcal{E}}$ ($\tilde{g}^{-1} \sim_{\tilde{\mathcal{E}}} g^{-1}$).*

To prove the above theorem, we follow a similar strategy as in Theorem 1 and establish $\tilde{\mathcal{G}}_{\mathsf{id}} = \tilde{\mathcal{G}}_{\mathsf{eq}}$. The proof of the above is in Section A.5 of the Appendix. Equivariances and imitators play similar roles in the way that they relax constraints on the encoder $\tilde{g}^{-1}$—any bijection that commutes with

either is a source of non-identifiability—but they are different from the perspective of how we should think about designing representation learning algorithms. Recall that $\mathcal{M}$ is composed of two sets of mechanisms, $\mathcal{M} = \mathcal{M}^* \cup \mathcal{M}'$, mechanisms in $\mathcal{M}^*$ that are used at least once in the evolution of the environment and those mechanisms in $\mathcal{M}'$ which are hypothesized but never used. Equivariances are dictated only by $\mathcal{M}^*$, which characterizes the evolution of the environment. Any increases to the number of distinct mechanisms in $\mathcal{M}^*$ will potentially decrease the number of equivariances shared by all mechanisms. This can only be achieved by modifying the environment in some way, either through an explicit intervention that modifies its mechanisms or by collecting data from multiple environments with diverse set of mechanisms. For example, in the bouncing balls example given in Figure 1, one could change the environment by varying the mass of the balls or observing it under different gravity conditions; or one could intervene by, say, changing the shape of balls in the system such that you get a different bouncing mechanism.

Imitators, by contrast, are a function of both the mechanisms $\mathcal{M}^*$ that were used and those that were hypothesized, $\mathcal{M}'$. An imitator is just some mechanism that can imitate another via some bijection, so one would expect that as we grow the number of mechanisms in $\mathcal{M}$, the size of the set of imitators can only grow; but interestingly, this is not always the case for mechanisms from $\mathcal{M}^*$. Recall that any $a \in \tilde{\mathcal{E}}$ produces an encoder of the form $\tilde{g}^{-1} = a \circ g^{-1}$, so the same transformation has to be used for all imitations and equivariances among the mechanisms in $\mathcal{M}^*$. Because of this, it is possible that increasing the size of $\mathcal{M}^*$ reduces the number of imitators. For example, if there is some mechanism, $m_i$, that does not commute with any non-trivial $a$ in $\tilde{\mathcal{E}}$ (either by imitation or equivariance), then adding $m_i$ to $\mathcal{M}^*$ will make the problem exactly identified. On the other hand, growing the size of the set of unused mechanisms, $\mathcal{M}'$, can result in significant non-identifiability. For example, if $\mathcal{M}$ consisted of a flexible class of functions (e.g. a multi-layer perceptron) then it would be easy to construct imitators of the form $m_1 = a^{-1} \circ m_2 \circ a$.

**Illustrating Theorem 3.** We consider the same setting as in Theorem 2. For each $t \in \{1, \cdots, d+1\}$, $m(z_t) = M z_t + b_t$ and $x_t = g(z_t)$. We only know that the mechanism is affine and only the offset $b_t$ is changing, but parameters $M$ and $b_t$ are not known. Let us construct the set $\tilde{\mathcal{E}}$ corresponding to the above setting. We can show that the set $\tilde{\mathcal{E}}$ consists of affine functions (See the Appendix A.6 for details). From Theorem 3, we can thus conclude that for this data generation process even with *very little knowledge of the mechanism, we get linear identifiability*. This is weaker than the offset based identifiability in Theorem 2, but there we were required to know the entire affine mechanism.

## 3 STOCHASTIC MECHANISMS

The results thus far relied on the assumption that the evolution of a system could be described in terms of deterministic mechanisms. This deterministic approach models settings where the full latent state is observable (via some unknown encoder $g^{-1}$) at a short enough time interval that there is no uncertainty about the system's evolution. To generalize to cases where there is some uncertainty about the latent state's evolution, we now develop analogous identification results for stochastic mechanisms that induce conditional distributions over latent states. The systems evolves as

$$X_t \leftarrow g(Z_t), \quad Z_{t+1} \leftarrow m_t(Z_t, U_t), \tag{7}$$

where each $U_t$ is noise with each component sampled independently from standard uniform distribution Uniform$[0, 1]$, $Z_1 \sim \mathbb{P}_Z$, $m_t : \mathcal{Z} \times [0, 1]^d \to \mathcal{Z}$, and the decoder $g : \mathcal{Z} \to \mathcal{X}$ is a diffeomorphism (i.e. a smooth bijection with an invertible Jacobian, see definition A.1 (Kass & Vos, 2011)).

In this section, we will focus on the case where the true mechanism is unknown; the case where mechanism is known is a special case with no imitator, i.e., $\tilde{m}_t = m_t$ for all $t$. The goal is to search for an encoder $\tilde{g}^{-1}$, which is a diffeomorphism, that generates $\hat{X}_{t+1} = \tilde{g} \circ \tilde{m}_t(\tilde{g}^{-1}(x_t), \hat{U}_t)$, where $\hat{U}_t$ is a random vector with each component from Uniform$[0, 1]$. In the deterministic case, we had required that any candidate encoder, $\tilde{g}^{-1}$, was point-wise consistent with the pairs of observations $(x_{t+1}, x_t)$. Here, encoders are only required to match the observed conditional distributions. An encoder that is consistent with the observed data can be used to generate $\hat{X}_{t+1}$ such that the distribution of $\hat{X}_{t+1} | X_t = x_t$ matches the distribution of $X_{t+1} | X_t = x_t$ for all $x_t \in \mathcal{X}$,

$$X_{t+1} | X_t = x_t \stackrel{d}{=} \hat{X}_{t+1} | X_t = x_t$$

$$g \circ m_t(g^{-1}(x_t), U_t) \stackrel{d}{=} \tilde{g} \circ \tilde{m}_t(\tilde{g}^{-1}(x_t), \hat{U}_t) \tag{8}$$

Now, define the set of all candidate decoders $\tilde{g}$ (with corresponding encoders $\tilde{g}^{-1}$) that solve the above equation 8 as $\mathcal{G}^s_{id}$. We can extend the notion of equivariance and imitation to the stochastic case by replacing equality in value by equality in distribution, and show that, as before, these are the only sources of non-identifiability. In the special case where $m_t$ is known, $\tilde{\mathcal{E}}^s$, defined below, only contains maps that result from equivariance. We continue to use the $\mathcal{M}^*$ – set of mechanisms that are used at least once in the evolution of the system and $\mathcal{M}$ – hypothesis class of all the mechanisms.

**Definition 3.** *Equivariance and imitators in distribution w.r.t* $\mathcal{M}$. *Define a set of functions $\mathcal{E}^s$ that satisfy commutativity w.r.t the set of distributions $\mathcal{M}$ in the following sense. The set $\mathcal{E}^s$ comprises all the diffeomorphisms $a$ that satisfy the following condition: $a \in \mathcal{E}^s$ if and only if for each $m \in \mathcal{M}^*$, $\exists\, m' \in \mathcal{M}$ such that for all $z \in \mathcal{Z}$, $a \circ m(z, U) \stackrel{d}{=} m'(a(z), U)$, where each component of $U$ is sampled independently from Uniform$[0, 1]$ .*

Define the set of maps that identify true $g$ up to $\mathcal{E}^s$ as $\mathcal{G}^s_{eq} = \{\tilde{g} | \tilde{g} = g \circ a^{-1}, a \in \mathcal{E}^s\}$

**Theorem 4.** *If the data generation process follows equation 7, then the set of encoders that solve stochastic observation identity equation 8 identify the true $g^{-1}$ up to the equivariances and imitators in distribution w.r.t $\mathcal{M}$, $\mathcal{E}^s(\tilde{g}^{-1} \sim_{\mathcal{E}^s} g^{-1})$.*

To prove the above theorem, we follow a similar strategy as in Theorem 1 and establish $\mathcal{G}^s_{id} = \mathcal{G}^s_{eq}$. The proof of the above is provided in Section A.7 in the Appendix. Theorem 4 is consistent with the results that we developed for the deterministic case (Theorem 3), but because the mechanism is allowed to be stochastic, it allows us to generalize known results based on distributional assumptions; we give an example of this in the next section.

## 4    A MECHANISM-BASED PERSPECTIVE ON EXISTING RESULTS

Our primary motivation for understanding how mechanistic knowledge can aid identification, is to develop methods that do not require independence assumptions over the latent variables. However, independence assumptions are not incompatible with the mechanism-based perspective: they simply define a particular kind of mechanism, which then implies identification up to the mechanism's associated equivariances and imitators. We demonstrate this by re-deriving recent identification results from Klindt et al. (2020) using the mechanisms implied by their respective distributional assumptions. We begin by describing the data generation process used by Klindt et al. (2020) as a stochastic mechanism of the form of equation 7. For each $t \in \{1, \cdots, T\}$

$$Z_{t+1} = Z_t + V_t, \quad V_t = f(U_t), \quad U_t \sim \text{Uniform}[0, 1]^d \tag{9}$$

where $f$ is an inverse CDF such that each component of $V_t \in \mathbb{R}^d$ is sampled from the generalized Laplace distribution centred at zero with norm parameter $\alpha \neq 2$, $Z_1 \sim \mathbb{P}_Z$. Next, we want to use Theorem 4 to derive all the solutions to the observation identity in equation 8.

**Theorem 5.** *If the data generation process follows equation 9, and $\mathbb{P}_Z$ and the parameters defining $f$ are known (same assumption as in Klindt et al. (2020)), then the solution to the stochastic observation identity equation 8 leads to identifying the true representations up to permutation, sign-flip and offset.*

The proof is given in Section A.8 in the Appendix. The above theorem generalizes Theorem 1 from Klindt et al. (2020) as we do not require $\alpha < 2$ and rather we work with $\alpha \neq 2$. Alternatively, analogous results to those in Klindt et al. (2020); Hyvarinen & Morioka (2017) can be derived in settings where we do not know the distribution of $V_t$ but instead assume that we observe $X_t$ often enough that the difference between $Z_t$ and $Z_{t+1}$ is small. In particular, suppose that the data generation process follows equation 9 except each component of $V_t$ is an i.i.d. draw from a non-Gaussian with zero mean and $|V_t| < \delta$. Then as $\delta \to 0$ the true latent is identified up to permutation, sign-flip and offset. See Section A.9 for details.

## 5    RELATED WORKS

Non-linear independent component analysis (ICA) is a highly unidentified problem; several works (Hyvärinen & Pajunen, 1999; Locatello et al., 2019) have shown that it is impossible to invert the data generation process without placing restrictions on the data and models. In recent years, a lot of progress has been made on the problem of non-linear identification. Hyvarinen & Morioka (2016;

2017) provide the first proofs for identification in non-linear ICA. Hyvarinen & Morioka (2016) showed that if the latent variables are mutually independent, with each component evolving in time following a non-stationary time series without temporal dependence, then non-linear identification is possible. Hyvarinen & Morioka (2017) showed that non-linear identification is also possible if the latent variables are mutually independent, with each component evolving in time following a stationary time series with temporal dependence. In Hälvä & Hyvarinen (2020), the authors combine non-stationarity (Hyvarinen & Morioka, 2016) and temporal dependency (Hyvarinen & Morioka, 2017) and extend identifiability guarantees in somewhat more general settings. Khemakhem et al. (2020a); Hyvarinen et al. (2019); Khemakhem et al. (2020b) further generalized the previous results; in these works instead of using time the authors require observation of auxiliary information. Klindt et al. (2020) departs from other non-linear ICA works as it explicitly exploits the sparsity in the transitions of the latent variables (further details on Klindt et al. (2020) can be found in the previous section.). In Zimmermann et al. (2021), the authors show that minimizing contrastive losses commonly used in self-supervised learning can also guarantee identification provided the data (contrastive pairs) follow a specific choice of data generation process (e.g., contrastive pair is generated from a von Mises-Fisher distribution). In another line of work Locatello et al. (2020); Shu et al. (2019), the authors study the role of weak supervision in assisting disetanglement. In a recent work, Gresele et al. (2021), propose to add new form of constraints to non-linear ICA. The constraint is based on the observation that the decoder $g$ that gives rise to the image $x$ is composed of simpler functions that are mutually algorithmically independent; the authors exploit this inductive bias on the structure of $g$ to invert the data generation process. In another recent work von Kügelgen et al. (2021), the authors consider models where the latent variables are divided into two blocks – content and style block, where the latents are not necessarily independent. Using data augmentation on the style latents, the authors show block-wise identification for content block. In Section 4 we argued the existing distributional assumptions could be interpreted as particular choices of stochastic mechanisms; for more details, see Table 1 in the Appendix, where we describe the form of the respective mechanisms. In short, prior work has focused on identification guarantees under assumptions on the dependence between the different random variables, which is in sharp contrast to our approach, which focuses on identification under varying degrees of the knowledge of mechanisms that govern latent dynamics.

**Equivariance** There is significant recent interest in leveraging equivariance assumptions to design more efficient deep network architectures; for a recent survey, see Bronstein et al. (2021). The general recipe of this line of work is to design functions (deep network architectures) that enforce equivariances. Our setting inverts this recipe, in that we have some known function, $m(z)$, and we are interested in finding all of its equivariances. The relationship between distributions and their equivariances has a long history in the statistics literature (see e.g. Eaton, 1989, and the references therein). Our characterization of stochastic equivariances was inspired by the functional representations given in Bloem-Reddy & Teh (2020). Finally, the importance of group symmetries in representation learning was discussed in Higgins et al. (2018). Higgins et al. focus on the relationship between the symmetries of the environment and a model's representations, whereas we focus on how symmetries in the environment's transition function constrain the representation. The two perspectives are complementary, and in future work we hope to unify them.

## 6 DISCUSSION AND FUTURE WORK

This paper has presented the first systematic study of how mechanisms governing the dynamics of high-level variables can be used to identify these variables from low-level observations, and up to what equivariances, which depend on the mechanisms. We show that this perspective is both powerful—yielding significant constraints in the space of possible representation—and that it generalizes many known approaches. Moving forward, a natural direction is to build methods founded on this theory. We describe two natural losses based on the identity $\tilde{g} \circ m \circ \tilde{g}^{-1}(x_t) = x_{t+1}$. i) **Loss using observations:** We minimize next observation prediction error $\min_{\tilde{g} \in \mathcal{H}, \tilde{h} \in \mathcal{H}, \tilde{m} \in \mathcal{M}} \sum_t \mathbb{E}\left[ \|X_{t+1} - \tilde{g} \circ \tilde{m} \circ \tilde{h}(X_t)\|^2 \right]$ where $\mathcal{H}, \mathcal{M}$ is the hypothesis class of functions for decoder and mechanisms respectively. ii) **Loss using latents:** Alternatively, one could re-write the identity as $g^{-1} \circ x_{t+1} = m \circ g^{-1} \circ x_t$ and use square error or contrastive loss given as $\min_{\tilde{g} \in \mathcal{H}, \tilde{m} \in \mathcal{M}} \sum_t \mathbb{E}\left[ -\log\left( \frac{\tilde{g}(X_{t+1})^\mathsf{T} \tilde{m}\tilde{g}(X_t)}{\tilde{g}(X_{t+1})^\mathsf{T} \tilde{m}\tilde{g}(X_t) + \sum_\tau \tilde{g}(X_\tau)^\mathsf{T} \tilde{m}\tilde{g}(X_t)} \right) \right]$ where $\tau$ represents other time instances, i.e., $\tau \neq t + 1$. The positive pair in the contrastive loss is formed by the adjacent time instances and the negative pair is formed by non-adjacent time instances. In Appendix Section A.12 we share our initial results for the contrastive approach applied to 3d identification dataset (Zimmermann et al., 2021) .

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

| Approach | Assumptions |
|---|---|
| Time contrastive (Hyvarinen & Morioka, 2016) | $Z_t \leftarrow U_t$, each $U_t^j$ is independent and non-stationary |
| Permutation contrastive (Hyvarinen & Morioka, 2017) | Each $Z_{t+1}^j \leftarrow m(Z_t^j, U_t^j)$, stationary and not quasi-Gaussian |
| Slow VAE (Klindt et al., 2020) | $Z_{t+1} \leftarrow Z_t + f(U_t)$, each $f(U_t^i)$ is independent and generalized Laplace |
| Conditional VAE (Khemakhem et al., 2020a; Hyvarinen et al., 2019) | $Z \leftarrow m(O, U)$, all components of $Z$ are independent conditional on $O$ |
| Independently modulated component analysis (Khemakhem et al., 2020b) | $Z \leftarrow m(O, U)$, $m$ has a special structures allowing to relax conditional independence |
| Contrastive learning (Zimmermann et al., 2021) | $\tilde{Z} \leftarrow m(Z, U)$, $m$ is such that $\tilde{Z} \sim$ conditional von Mises-Fisher |
| Multi-view ICA (Gresele et al., 2020) | $Z_1 \leftarrow m(Z_0, U)$, $X_0 \leftarrow g_0(Z_0)$, $X_1 \leftarrow g_1(Z_1)$, all components of $Z_0$ and $U$ are independent |

Table 1: Table comparing different works and the assumptions made for identifiability.

# A APPENDIX

## A.1 PROOF OF THEOREM 1

*Proof.* First we show that $\mathcal{G}_{\text{id}} \subseteq \mathcal{G}_{\text{eq}}$.

Consider a $\tilde{g} \in \mathcal{G}_{\text{id}}$. For each $x \in \mathcal{X}$

$$g \circ m \circ g^{-1}(x) = \tilde{g} \circ m \circ \tilde{g}^{-1}(x)$$

$$\tilde{g}^{-1} \circ \left( g \circ m \circ g^{-1}(x) \right) = \tilde{g}^{-1} \circ \left( \tilde{g} \circ m \circ \tilde{g}^{-1}(x) \right) \qquad \text{(Compose } \tilde{g}^{-1} \text{ with both sides)}$$

$$\tilde{g}^{-1} \circ \left( g \circ m \circ g^{-1}(x) \right) = m \circ \tilde{g}^{-1}(x) \qquad (\tilde{g}^{-1} \circ \tilde{g}(z) = z)$$

Since $g$ is invertible we can substitute $x$ in the above equation with $x = g(z)$ and obtain for each $z \in \mathcal{Z}$

$$\left( \tilde{g}^{-1} \circ g \right) \circ m \circ \left( g^{-1} \circ g(z) \right) = m \circ \left( \tilde{g}^{-1} \circ g(z) \right)$$

$$\left( \tilde{g}^{-1} \circ g \right) \circ m(z) = m \circ \left( \tilde{g}^{-1} \circ g \right)(z) \qquad (10)$$

Define $\tilde{g}^{-1} \circ g = a$. Observe that $a$ is invertible and from equation 10 we gather that $a \in \mathcal{E}$. Also, since $\tilde{g} = g \circ a^{-1}$, we can conclude that $\tilde{g} \in \mathcal{G}_{\text{eq}}$, which proves the first part of the claim. For the second part, we show that $\mathcal{G}_{\text{eq}} \subseteq \mathcal{G}_{\text{id}}$.

Consider a $\tilde{g} \in \mathcal{G}_{\text{eq}} = \{ \tilde{g} \mid \tilde{g} = g \circ a^{-1}, \ a \in \mathcal{E} \}$. By definition, can express $\tilde{g} = g \circ a^{-1}$. For each $x \in \mathcal{X}$ we write

$$\tilde{g} \circ m \circ \tilde{g}^{-1}(x) = \left( g \circ a^{-1} \right) \circ m \circ \left( g \circ a^{-1} \right)^{-1}(x)$$

$$= \left( g \circ a^{-1} \right) \circ m \circ \left( a \circ g^{-1} \right)(x)$$

$$= \left( g \circ a^{-1} \right) \circ a \circ m \circ g^{-1}(x)$$

$$= g \circ m \circ g^{-1}(x)$$

Observe that $\tilde{g}$ is both a bijection and satisfies the identity in equation 4. Therefore, $\tilde{g} \in \mathcal{G}_{\text{id}}$. This proves the second part of the claim. Therefore, $\mathcal{G}_{\text{id}} = \mathcal{G}_{\text{eq}}$. $\qquad\square$

## A.2 LINEAR MECHANISM $M$ AND LINEAR $G$.

The focus of the main text is on nonlinear identification, but in this section we show how the analysis for general functions $g$ and $m$ applies to the special case when $g$ and $m$ are linear and affine maps respectively. This special case is useful both as a concrete example, and as a stepping stone to

Theorem 2 (nonlinear $g$ with affine $m$) which will reuse some of the same proof strategies. We write the data generation process as follows.

$$
\begin{aligned}
z_{t+1} &= M z_t + b, \\
x_t &= G z_t,
\end{aligned}
\tag{11}
$$

where $M \in \mathbb{R}^{d \times d}$ is a diagonalizable matrix describing the mechanism, $b \in \mathbb{R}^d$ is the offset parameter, $G \in \mathbb{R}^{d \times d}$ is an invertible matrix determining how the data transforms from latent space to the observable space. We write the eigendecomposition of $M$ as follows $M = S\Lambda S^{-1}$, where $S$ is the matrix of eigenvectors and $\Lambda$ is a diagonal matrix of the eigenvalues. On the same lines as the equation 2, we can obtain an identity between $x_t$ and $x_{t+1}$ as follows.

$$
\begin{aligned}
z_{t+1} &= M z_t + b \\
G^{-1} x_{t+1} &= M G^{-1} x_t + b \\
x_{t+1} &= G M G^{-1} x_t + G b
\end{aligned}
\tag{12}
$$

If the learner knows $M$ and $b$, it tries to solve for an invertible $\tilde{G}$ that satisfies

$$
x_{t+1} = \tilde{G} M \tilde{G}^{-1} x_t + \tilde{G} b
\tag{13}
$$

**Theorem 6.** *If the data generation process follows equation 11 and the eigenvalues of the mechanism $M$ are all distinct and each component of the vector $S^{-1}b$ is non-zero, then the only solution to the identity in equation 13 is the true mechanism $G$.*

*Proof.* We take the difference of the equations 12 and 13 to get the following condition. For each $x_t \in \mathbb{R}^d$

$$
(G M G^{-1} - \tilde{G} M \tilde{G}^{-1}) x_t + (G - \tilde{G}) b = 0
\tag{14}
$$

Because, equations 12 and 13 hold for all $x_t$, we can substitute $x_t = 0$ in the above to get

$$
(G - \tilde{G}) b = 0
\tag{15}
$$

We plug the above condition in equation 15 back into equation 14 to get the following condition. For each $x_t \in \mathbb{R}^d$

$$
(G M G^{-1} - \tilde{G} M \tilde{G}^{-1}) x_t = 0
\tag{16}
$$

If equation 16 holds for $d$ linearly independent vectors $x_t \in \mathbb{R}^d$, then we can conclude that,

$$
\begin{aligned}
G M G^{-1} - \tilde{G} M \tilde{G}^{-1} &= 0 \\
G^{-1}\left(G M G^{-1} - \tilde{G} M \tilde{G}^{-1}\right) &= 0 \\
M G^{-1} - G^{-1} \tilde{G} M \tilde{G}^{-1} &= 0 \\
\left(M G^{-1} - G^{-1} \tilde{G} M \tilde{G}^{-1}\right) \tilde{G} &= 0 \\
M G^{-1} \tilde{G} &= G^{-1} \tilde{G} M
\end{aligned}
\tag{17}
$$

Let $A = G^{-1} \tilde{G}$. We substitute $A$ and the eigendecomposition of $M$ ($M = S\Lambda S^{-1}$, where $\Lambda = \text{diag}(\lambda_1, \cdots, \lambda_d)$) in equation 17 to get

$$
\begin{aligned}
M &= A M A^{-1} \\
S\Lambda S^{-1} &= A S\Lambda S^{-1} A^{-1} \\
\Lambda &= \left(S^{-1} A S\right)\Lambda\left(S^{-1} A^{-1} S\right) \\
\Lambda &= C\Lambda C^{-1} \qquad \text{(where } C = S^{-1} A S) \\
\Lambda C &= C\Lambda
\end{aligned}
\tag{18}
$$

We further simplify equation 18 and compare each element of the matrix $\Lambda C$ and $C\Lambda$ to get the following condition. For all $i, j \in \{1, \cdots, d\}$

$$[\Lambda C]_{ij} = C_{ij}\lambda_i, \quad [C\Lambda]_{ij} = C_{ij}\lambda_j$$
$$C_{ij}(\lambda_i - \lambda_j) = 0 \tag{19}$$

Consider the above equation 19 for $i \neq j$ to get

$$C_{ij}(\lambda_i - \lambda_j) = 0 \implies C_{ij} = 0 \text{ (we use the assumption that } \lambda_i \neq \lambda_j) \tag{20}$$

From the above, it follows that $C$ is a diagonal matrix. We obtain an expression for $A$ in terms of $C$ matrix below.

$$S^{-1}AS = C$$
$$A = SCS^{-1} \tag{21}$$

From equation 15 we get that

$$(G - \tilde{G})b = 0 \overset{\tilde{G}=GA}{\implies} (I - A)b = 0 \overset{\text{Eqn. (21)}}{\implies} (I - SCS^{-1})b = 0$$
$$\underbrace{\implies}_{\text{Left multiply } S^{-1}} S^{-1}b - CS^{-1}b \implies (C - I)S^{-1}b = 0 \tag{22}$$

Since $C$ is a diagonal matrix, we can simplify the above condition further to get $(c_{ii} - 1)(S^{-1}b)_i = 0$. Since $(S^{-1}b)_i \neq 0 \implies c_{ii} = 1$, we obtain $C = I \implies G^{-1}\tilde{G} = I \implies G = \tilde{G}$.

$\square$

## A.3 Proof of Corollary 1

*Proof.* This proof follows essentially the same strategy as the proof of Theorem 1 with a set of mechanisms, $\mathcal{M}^*$ instead of a single mechanism $m$; we include it for completeness. First we show that $\mathcal{G}^*_{\text{id}} \subseteq \mathcal{G}^*_{\text{eq}}$.

Consider a $\tilde{g} \in \mathcal{G}^*_{\text{id}}$. For each $x \in \mathcal{X}$ and for each $m \in \mathcal{M}^*$

$$g \circ m \circ g^{-1}(x) = \tilde{g} \circ m \circ \tilde{g}^{-1}(x), \tag{23}$$

and by following the same steps as the proof of Theorem 1, we can show that,

$$\left(\tilde{g}^{-1} \circ g\right) \circ m(z) = m \circ \left(\tilde{g}^{-1} \circ g\right)(z) \tag{24}$$

As before, define $\tilde{g}^{-1} \circ g = a$. Observe that $a$ is invertible and from equation 24 we gather that $a \in \mathcal{E}^*$. Also, since $\tilde{g} = g \circ a^{-1}$, we can conclude that $\tilde{g} \in \mathcal{G}^*_{\text{eq}}$, which proves the first part of the claim. In the second part, we need to show that $\mathcal{G}^*_{\text{eq}} \subseteq \mathcal{G}^*_{\text{id}}$.

Consider a $\tilde{g} \in \mathcal{G}^*_{\text{eq}} = \{\tilde{g} \mid \tilde{g} = g \circ a^{-1}, \ a \in \mathcal{E}^*\}$. By definition, can express $\tilde{g} = g \circ a^{-1}$. For each $x \in \mathcal{X}$ and for each $m \in \mathcal{M}^*$ we write,

$$\tilde{g} \circ m \circ \tilde{g}^{-1}(x) = \left(g \circ a^{-1}\right) \circ m \circ \left(a \circ g^{-1}\right)(x) =$$
$$\left(g \circ a^{-1}\right) \circ a \circ m\left(g^{-1}(x)\right) \quad \text{(since } a \text{ commutes with each } m \in \mathcal{M}) = \tag{25}$$
$$g \circ m \circ g^{-1}(x) \quad \text{for all } m$$

Observe that $\tilde{g}$ is both a bijection and satisfies the observation identity in equation 4. Therefore, $\tilde{g} \in \mathcal{G}^*_{\text{id}}$. This proves the second part of the claim. Therefore, $\mathcal{G}^*_{\text{id}} = \mathcal{G}^*_{\text{eq}}$.

$\square$

## A.4 Proof of Theorem 2

*Proof.* We showed in Theorem 1 that the only source of non-identifiability are the bijections, $a$, in the set $\mathcal{E}$; our task here is to explicitly find all of these bijections for affine mechanisms. If $a \in \mathcal{E}$, then it satisfies $a \circ m = m \circ a$. We replace $m$ with the affine mechanism to obtain the following condition. For each $z \in \mathbb{R}^d$

$$a(Mz + b) = Ma(z) + b \tag{26}$$

Next, recall that $a : R^d \to R^d$; we take gradient of the function in the LHS and RHS of the above equation 26 separately w.r.t $z$. Consider the $j^{th}$ component of $a(Mz + b)$ denoted as $a_j(Mz + b)$. We first take the gradient of $a_j(Mz + b)$ w.r.t $z$

$$\nabla_z a_j(Mz + b) = \left(\frac{dy}{dz}\right)^{\mathsf{T}} \nabla_y a_j(y), \tag{27}$$

where $y = Mz + b$, $\nabla_y a_j(y)$ is the gradient of $a_j$ w.r.t $y$ and $\frac{dy}{dz}$ denotes the Jacobian of $y$ w.r.t $z$. We simplify the above further to get

$$\nabla_z a_j(Mz + b) = M^{\mathsf{T}} \nabla_y a_j(y) = M^{\mathsf{T}} \nabla_y a_j(Mz + b) \tag{28}$$

We can write the above for each component of $a$ as follows.

$$
\begin{aligned}
\left[\nabla_z a_1(Mz + b), \cdots, \nabla_z a_d(Mz + b)\right] &= \left[M^{\mathsf{T}} \nabla_y a_1(Mz + b), \cdots, M^{\mathsf{T}} \nabla_y a_d(Mz + b)\right] \\
&= M^{\mathsf{T}}\left[\nabla_y a_1(Mz + b), \cdots, \nabla_y a_d(Mz + b)\right] = M^{\mathsf{T}} J^{\mathsf{T}}(Mz + b),
\end{aligned} \tag{29}
$$

where $J(Mz + b)$ is the Jacobian of $a$ computed at $Mz + b$. Next, we take the gradient of the $j^{th}$ component of the RHS in equation 26 and let $m_j$ denote the $j^{\text{th}}$ column of $M$,

$$\nabla_z\left[m_j^{\mathsf{T}} a(z) + b_j\right] = \sum_i m_{ji} \nabla_z a_i(z) = \left[\nabla a_1(z), \cdots, \nabla a_d(z)\right] \begin{bmatrix} m_{j1} \\ m_{j2} \\ \vdots \\ m_{jd} \end{bmatrix} \tag{30}$$

We can write the above for each component in the RHS of equation 26 as follows

$$\left[\nabla_z\left[m_1^{\mathsf{T}} a(z) + b_1\right], \cdots, \nabla_z\left[m_d^{\mathsf{T}} a(z) + b_d\right]\right] = \left[\nabla a_1(z), \cdots, \nabla a_d(z)\right] \begin{bmatrix} m_{11}, \cdots, m_{d1} \\ m_{12}, \cdots, m_{d2} \\ \vdots \qquad \vdots \\ m_{1d}, \cdots, m_{dd} \end{bmatrix} = J^{\mathsf{T}}(z) M^{\mathsf{T}} \tag{31}$$

We equate the gradient of LHS and RHS in equation 26 using the expressions derived in equation 29 and equation 31 to obtain

$$a(Mz + b) = Ma(z) + b \implies M^{\mathsf{T}} J^{\mathsf{T}}(Mz + b) - J^{\mathsf{T}}(z) M^{\mathsf{T}} = 0 \tag{32}$$

We write the same expression at another offset $b' \neq b$ below

$$a(Mz + b') = Ma(z) + b \implies M^{\mathsf{T}} J^{\mathsf{T}}(Mz + b') - J^{\mathsf{T}}(z) M^{\mathsf{T}} = 0 \tag{33}$$

Taking the difference of equation 32 and equation 33 we get $M^{\mathsf{T}} J^{\mathsf{T}}(Mz + b) = M^{\mathsf{T}} J^{\mathsf{T}}(Mz + b')$. Since $M$ is invertible, we get $J(Mz + b) = J(Mz + b')$. Consider row $j$ of this identity. For each $z \in \mathbb{R}^d$

$$\nabla a_j(Mz+b) - \nabla a_j(Mz+b') = 0 \implies \nabla a_j(\tilde{z}) - \nabla a_j(\tilde{z}+b'-b) = 0 \implies \begin{bmatrix} \nabla_1^2 a_j(\theta_1) \\ \nabla_2^2 a_j(\theta_2) \\ \vdots \\ \nabla_d^2 a_j(\theta_d) \end{bmatrix}(b-b') = 0 \tag{34}$$

where $\nabla^2 a_j$ is the Hessian of $a_j$ and $\nabla_k^2 a_j(\theta_k)$ corresponds to the $k^{th}$ row of the Hessian matrix. Note that in the above expansion there is a different $\theta_k$ for each row (mean value theorem applied

to each component of $\nabla a_j$ yields a different point $\theta_k$ on the line joinining $\tilde{z}$ and $\tilde{z} + b - b'$. From Assumption 2 and based on the fact that $M$ is invertible, it follows that $\nabla_k^2 a_j(\theta_k)(b - b') = 0$ over a measurable set. Since $a_j$ is analytic $\nabla_k^2 a_j(z)(b - b')$ is also analytic. Therefore, from (Mityagin, 2015), we can conclude that $\nabla_k^2 a_j(z)(b - b') = 0$ for all $z$. We can make the same argument for each component $k$ and conclude that $\nabla^2 a_j(z)(b - b') = 0$. From Assumption 1, it follows that $\nabla^2 a_j(z)(b^j - b^1) = 0$ for all $j \in \{2, \cdots, d+1\}$ and since the set $\{b^2 - b^1, \cdots, b^{d+1} - b^1\}$ is linearly independent $\nabla^2 a_j(z) = 0$ for all $z$. This implies $a(z) = Az + p$. Plug $a(z) = Az + p$ into $a(Mz + b) = Ma(z) + b$ to get

$$A(Mz + b) + p = MA(z + p) + b \implies (AM - MA)z + (A - I)b + (I - M)p = 0 \quad (35)$$

We write the same expression for offset $b'$

$$A(Mz + b') + p = MA(z + p) + b' \implies (AM - MA)z + (A - I)b' + (I - M)p = 0 \quad (36)$$

We take the difference of equation 35 and equation 36 to get

$$(A - I)(b - b') = 0 \quad (37)$$

Substitute $z = 0$ in equation 35 to get

$$(A - I)b + (I - M)p = 0 \quad (38)$$

Substitute the above condition in equation 38 into equation 35 to get the following. For each $z$

$$(AM - MA)z = 0 \quad (39)$$

We can now leverage the proofs from the linear setting in Section A.2. The above equation 39 is the same as equation 17 and the equation 37 is the same as equation 22, with $b$ replaced by $b - b'$. Following the same analysis as before, we get that latent variables are exactly identified; we show all the steps below for completeness. By choosing $d$ linearly independent $z$ and substituting in equation 39 we get the following,

$$\begin{aligned}
MA &= AM \\
M &= AMA^{-1}, \\
S\Lambda S^{-1} &= AS\Lambda S^{-1}A^{-1}, \\
\Lambda &= \left(S^{-1}AS\right)\Lambda\left(S^{-1}A^{-1}S\right), \\
\Lambda &= C\Lambda C^{-1}, \\
\Lambda C &= C\Lambda,
\end{aligned} \quad (40)$$

where $C = S^{-1}AS$, $M = S\Lambda S^{-1}$, $\Lambda = \text{diag}\left(\lambda_1, \cdots, \lambda_d\right)$. We further simplify equation 18 and compare each element of the matrix $\Lambda C$ and $C\Lambda$ to get the following condition. For all $i, j \in \{1, \cdots, d\}$

$$\begin{aligned}
[\Lambda C]_{ij} &= C_{ij}\lambda_i, \quad [C\Lambda]_{ij} = C_{ij}\lambda_j \\
C_{ij}(\lambda_i - \lambda_j) &= 0
\end{aligned} \quad (41)$$

Consider the above equation 41 for $i \neq j$ to get

$$C_{ij}(\lambda_i - \lambda_j) = 0 \implies C_{ij} = 0 \text{ (we use the assumption that } \lambda_i \neq \lambda_j) \quad (42)$$

From the above, it follows that $C$ is a diagonal matrix. We obtain an expression for $A$ in terms of $C$ matrix below.

$$\begin{aligned}
S^{-1}AS &= C \\
A &= SCS^{-1}
\end{aligned} \quad (43)$$

From equation 37 we get that

$$(G - \tilde{G})(b - b') = 0 \overset{\tilde{G}=GA}{\implies} (I - A)(b - b') = 0 \implies (I - SCS^{-1})b = 0$$
$$\underbrace{\implies}_{\text{Left multiply } S^{-1}} S^{-1}(b - b') - CS^{-1}(b - b') \implies (C - I)S^{-1}(b - b') = 0 \quad (44)$$

Since $C$ is a diagonal matrix, we can simplify the above condition further to get $(c_{ii} - 1)(S^{-1}(b - b'))_i = 0$. Since $(S^{-1}(b-b'))_i \neq 0 \implies c_{ii} = 1$, we obtain $C = I \implies A = I \implies a(z) = z+p$. This proves that the latents are identified up to an offset. $\qquad \square$

## A.5 PROOF OF THEOREM 3

*Proof.* We first show that $\tilde{\mathcal{G}}_{id} \subseteq \tilde{\mathcal{G}}_{eq}$. Consider a $\tilde{g} \in \tilde{\mathcal{G}}_{id}$. We rewrite equation 6 below. For all $x \in \mathcal{X}$

$$
\begin{aligned}
g \circ m_t \circ g^{-1}(x) &= \tilde{g} \circ \tilde{m}_t \circ \tilde{g}^{-1}(x) \\
(\tilde{g}^{-1} \circ g) \circ (m_t \circ g^{-1}(x)) &= \tilde{m}_t \circ \tilde{g}^{-1}(x)
\end{aligned}
\tag{45}
$$

Since $g$ is bijective, we can write $x = g(z)$ to get

$$
(\tilde{g}^{-1} \circ g) \circ m_t(z) = \tilde{m}_t \circ \tilde{g}^{-1} \circ g(z)
\tag{46}
$$

Since the above equality holds for all $z \in \mathcal{Z}$ we can conclude that

$$
\begin{aligned}
(\tilde{g}^{-1} \circ g) \circ m_t &= \tilde{m}_t \circ (\tilde{g}^{-1} \circ g), \\
a \circ m_t &= \tilde{m}_t \circ a,
\end{aligned}
\tag{47}
$$

The above conclusion in equation 47 holds for all $m_t \in \mathcal{M}^*$. Therefore, $a$ in equation 47 and $\{\tilde{m}_t\}_{t=1}^T$ (where $\tilde{m}_t \in \mathcal{M}$) together satisfy the condition that for all $m \in \mathcal{M}^*$, $a \circ m = \tilde{m} \circ a$, where $\tilde{m} \in \mathcal{M}$. We can rewrite $\tilde{g}^{-1} \circ g = a$ as $\tilde{g} = g \circ a^{-1}$. From this it follows that $\tilde{g} \in \tilde{\mathcal{G}}_{eq}$. This proves the first part of the theorem.

Now let us consider the second part of the theorem. Consider a $\tilde{g} \in \tilde{\mathcal{G}}_{eq}$. We can write $\tilde{g} = g \circ a^{-1}$, where $a \in \tilde{\mathcal{E}}$. At time $t$, some mechanism $m_t \in \mathcal{M}^*$ is used to transform the latents. Since $a \in \tilde{\mathcal{E}}$, select the mechanism $\tilde{m}_t \in \mathcal{M}$ for which $a \circ m_t = \tilde{m}_t \circ a$ and as a consequence

$$
g \circ m_t \circ g^{-1} = g \circ a^{-1} \circ \tilde{m}_t \circ a \circ g^{-1} = \tilde{g} \circ \tilde{m}_t \circ \tilde{g}^{-1}
\tag{48}
$$

In the first equality above, we use $a \circ m_t = \tilde{m}_t \circ a$ and in the second equality we use the definition of $\tilde{g}$. We can repeat the above exercise for all $t$ and corresponding $m_t$ using the same $a$. Therefore, $\tilde{g}$ is in $\tilde{\mathcal{G}}_{id}$. This shows the second part of the theorem, i.e., $\tilde{\mathcal{G}}_{eq} \subseteq \tilde{\mathcal{G}}_{id}$.

$\square$

## A.6 LEVERAGING THEOREM 3 WHEN MECHANISM IS LINEAR AND $g$ IS NON-LINEAR

We write the data generation process as follows. For each $t \in \{1, \cdots, d+1\}$

$$
\begin{aligned}
z_{t+1} &= M_t z_t + b_t, \\
x_t &= g(z_t),
\end{aligned}
\tag{49}
$$

Let us construct the set $\tilde{\mathcal{E}}$ corresponding to the above setting. For each $z \in \mathbb{R}^d$,

$$
\begin{aligned}
a(Mz + b) = M' a(z) + \tilde{b} &\implies M^\mathsf{T} J^\mathsf{T}(Mz + b) - J^\mathsf{T}(z) M'^{,\mathsf{T}} = 0, \\
a(Mz + b') = M' a(z) + \tilde{b}' &\implies M^\mathsf{T} J^\mathsf{T}(Mz + b') - J^\mathsf{T}(z) M'^{,\mathsf{T}} = 0,
\end{aligned}
\tag{50}
$$

where $(M, b)$ and $(M, b')$ are the true mechanisms and $(M', b')$ and $(M', \tilde{b}')$ are the imtitating mechanisms chosen by the learner, $J$ is the Jacobian of $a$. Note here the learner only exploits the knowledge that $b$ changes to $\tilde{b}$, which is why it keeps $M'$ fixed and only changes the offset. We take the difference of the RHS in the above two equations to get

$$
M^\mathsf{T} J^\mathsf{T}(Mz + b) = M^\mathsf{T} J^\mathsf{T}(Mz + b')
\tag{51}
$$

Since $M$ is invertible we get $J(Mz + b) - J(Mz + b') = 0$ for all $z$. We can follow the same justification as was used in equation 33 to conclude that $J(z)$ is constant and $a$ is thus an affine map. We substitue the affine map $a(z) = Az + p$ back into equation 50 to get the following. For all $z$

$$
AMz + Ab^j + p = M' Az + b'^{,j} + M' p
\tag{52}
$$

Substitute $z = 0$ to get $b'^{,j} = Ab^j + p - M'p$. Substitute this condition back into the above equation, we get $AM = M'A \implies M' = AMA^{-1}$.

## A.7 Proof of Theorem 4

Before stating the proof of Theorem 4, we state two existing results that we use.

**Result 1. (Change of variables formula (DeGroot, 2012))** Given a continuous random variable $X \in \mathbb{R}^d$ with pdf $p_X$ and its transformation $Y = f(X)$, where $f : \mathbb{R}^d \to \mathbb{R}^d$ is a diffeomorphism,[8] then $p_Y(f(x))|\det(J_f(x))| = p_X(x)$, where $J_f$ is the Jacobian of $f$ computed at $x$.

**Lemma 1.** *If $X$ and $Y$ are two continuous random variables that take values in $\mathbb{R}^d$ that are equal in distribution, i.e., $X \overset{d}{=} Y$. If $f : \mathbb{R}^d \to \mathbb{R}^d$ is a diffeomorphism, then $f(X) \overset{d}{=} f(Y)$.*

*Proof.* Since $X$ and $Y$ are equal in distribution, they have the same pdfs, i.e. $p_X(x) = p_Y(x)$ for all $x \in \mathbb{R}^d$. We can use the change of variables formula in Resut 1 above to get the following. Let $W = f(X)$, $p_X(f^{-1}(w))|\det(J_{f^{-1}}(w))| = p_W(w)$ and let $V = f(Y)$, $p_Y(f^{-1}(v))|\det(J_{f^{-1}}(v))| = p_V(v)$. Comparing the two expressions when $w = v$ we get $p_W(w) = p_V(w)$. This proves the result. □

We stated Result 1 and Lemma 1 for continuous random variables. When the random variables are discrete, Lemma 1 holds for any function $f$.

**Lemma 2.** *(Kass & Vos, 2011) If $f : \mathcal{Z} \to \mathcal{Z}$ and $g : \mathcal{Z} \to \mathcal{Z}$ are diffeomorphisms, then $f \circ g$ is a diffeomorphism.*

*Proof.* We first show that $\mathcal{G}_{\text{id}}^{\text{s}} \subseteq \mathcal{G}_{\text{eq}}^{\text{s}}$.

From the observation identity in equation 8 we get that $\tilde{g}, \{\tilde{m}_t\}_{t=1}^T$ satisfy the following for all $t \in \{1, \cdots, T\}$

$$
\begin{aligned}
g \circ m_t\big(g^{-1}(x_t), U_t\big) &\overset{d}{=} \tilde{g} \circ \tilde{m}_t\big(\tilde{g}^{-1}(x_t), \hat{U}_t\big) \\
\big(\tilde{g}^{-1} \circ g\big) \circ m_t\big(g^{-1}(x_t), U_t\big) &\overset{d}{=} \tilde{m}_t\big(\tilde{g}^{-1}(x_t), \hat{U}_t\big)
\end{aligned}
\tag{53}
$$

In the second step in the above equation, we transformed the random variables in the first step using the same transform $\tilde{g}^{-1}$. $\tilde{g}^{-1}$ is a diffeomorphism; we compose both sides of the first step LHS and RHS with $\tilde{g}^{-1}$. We use Lemma 1 to get from the first step to the second step in the above equation equation 53. In the above equation $x_t$ is a fixed value and the only source of randomness is from $U_t$ in LHS and $\hat{U}_t$ in the RHS. We substitute $x_t = g(z_t)$ to further simplify the above expression in equation 53

$$
\begin{aligned}
\big(\tilde{g}^{-1} \circ g\big) \circ m_t\big(g^{-1} \circ g(z_t), U_t\big) &\overset{d}{=} \tilde{m}_t\big(\tilde{g}^{-1} \circ g(z_t), \hat{U}_t\big) \\
\big(\tilde{g}^{-1} \circ g\big) \circ m_t\big(z_t, u_t\big) &\overset{d}{=} \tilde{m}_t\big(\tilde{g}^{-1} \circ g(z_t), \hat{U}_t\big)
\end{aligned}
\tag{54}
$$

Substitute $a = \tilde{g}^{-1} \circ g$ in the above to get the following

$$
\begin{aligned}
a \circ m_t\big(z_t, U_t\big) &\overset{d}{=} \tilde{m}_t\big(a(z_t), \hat{U}_t\big) \\
a \circ m_t\big(z_t, U_t\big) &\overset{d}{=} \tilde{m}_t\big(a(z_t), U_t\big)
\end{aligned}
\tag{55}
$$

From Lemma 2 it follows that $a$ in the above is a diffeomorphism. Since we assume that $\cup_{t=1}^T \{m_t\} = \mathcal{M}^*$ it follows that $a$ in equation 47 and $\{\tilde{m}_t\}_{t=1}^T$ (where $\tilde{m}_t \in \mathcal{M}$) together satisfy the condition for membership in $\mathcal{E}^{\text{s}}$. Since $\tilde{g} = g \circ a^{-1}$ we obtain that $\tilde{g} \in \mathcal{G}_{\text{eq}}^{\text{s}}$.

We now show that $\mathcal{G}_{\text{eq}}^{\text{s}} \subseteq \mathcal{G}_{\text{id}}^{\text{s}}$. Consider a $\tilde{g} \in \mathcal{G}_{\text{eq}}^{\text{s}}$. We use $\tilde{g} = g \circ a^{-1}$, where $a \in \mathcal{E}^{\text{s}}$ to simplify the following random variable

---

[8] http://math.mit.edu/~larsh/teaching/F2007/handouts/changeofvariables.pdf

$$g \circ m_t(g^{-1}(x_t), U_t) = \left(g \circ a^{-1} \circ a\right) \circ m_t(g^{-1}(x_t), U_t) = \tilde{g} \circ a \circ m_t(g^{-1}(x_t), U_t) \qquad (56)$$

Since $a \in \mathcal{E}^{\mathsf{s}}$ we have

$$a \circ m_t(g^{-1}(t), U_t) \overset{d}{=} \tilde{m}_t\left(a \circ g^{-1}(x_t), U_t\right)$$

From Lemma 2 it follows that $\tilde{g}$ is a diffeomorphism. From Lemma 1 it follows that

$$\tilde{g} \circ a \circ m_t(g^{-1}(x_t), U_t) \overset{d}{=} \tilde{g} \circ \tilde{m}_t(a \circ g^{-1}(x_t), U_t) = \tilde{g} \circ \tilde{m}_t(\tilde{g}^{-1}(x_t), U_t) \qquad (57)$$

Combining equation 56 and equation 57 and using the fact that $\hat{U}_t \overset{d}{=} U_t$ we get

$$g \circ m_t(g^{-1}(x_t), U_t) \overset{d}{=} \tilde{g} \circ \tilde{m}_t(\tilde{g}^{-1}(x_t), U_t) \overset{d}{=} \tilde{g} \circ \tilde{m}_t(\tilde{g}^{-1}(x_t), \hat{U}_t) \qquad (58)$$

From the definition of $\mathcal{E}^{\mathsf{s}}$ it follows with the same choice of $a$ the condition continues to hold for all $m_t \in \mathcal{M}^*$. Therefore, $\tilde{g} \in \mathcal{G}_{\mathsf{id}}^{\mathsf{s}}$. This proves the second part of the theorem.

$\square$

## A.8   PROOF OF THEOREM 5

*Proof.* In Theorem 4, we showed that all the solutions to the observation identity in equation 8 can be characterized in terms of the equivariances in distribution defined by the set $\mathcal{E}^{\mathsf{s}}$. Let us analyze the set $\mathcal{E}^{\mathsf{s}}$ for the class of mechanisms considered in Klindt et al. (2020). Consider a $a \in \mathcal{E}^{\mathsf{s}}$. For each $z \in \mathbb{R}^d$

$$a(z + V) \overset{d}{=} a(z) + \hat{V}, \qquad (59)$$

Define $\hat{Y} = a(z) + \hat{V}$. Since $\hat{V} \overset{d}{=} V$ we write the probability density function (pdf) of $\hat{Y}$ as

$$f_{\hat{Y}}(y) = f_V(y - a(z)) \qquad (60)$$

Define $Y = a(z + V)$. $a : \mathbb{R}^d \to \mathbb{R}^d$ is a diffeomorphism. We use the change of variables result (Result 1) to write the pdf $Y$ as follows. For each $y \in \mathbb{R}^d$

$$f_Y(y) = \frac{1}{\left|\det\left(J\left(a^{-1}(y)\right)\right)\right|} f_V\left(a^{-1}(y) - z\right), \qquad (61)$$

where $J(a^{-1}(y))$ is the Jacobian of $a$ computed at $a^{-1}(y)$, and det is the determinant.

We substitute equation 60 and equation 61 in the equivariance condition in equation 59 to obtain the following. For each $y \in \mathbb{R}^d$

$$Y \overset{d}{=} \hat{Y}$$
$$f_V(y - a(z)) = \frac{f_V(a^{-1}(y) - z)}{|\det(J(a^{-1}(y)))|} \qquad (62)$$
$$f_V(a(w) - a(z)) = \frac{f_V(w - z)}{|\det(J(w))|},$$

where $w = a^{-1}(y)$. In the above we equated the conditionals for each $z$, we now equate the marginals.

$$g \circ (Z + V) \overset{d}{=} \tilde{g} \circ (\hat{Z} + \hat{V})$$
$$a \circ (Z + V) \overset{d}{=} \hat{Z} + \hat{V} \qquad (63)$$

We follow Klindt et al. (2020) and assume $\hat{Z} \overset{d}{=} Z$ and $\hat{V} \overset{d}{=} V$. Therefore, $Z + V \overset{d}{=} \hat{Z} + \hat{V}$. We use this condition to restate equation 63 as

$$a \circ (Z + V) \overset{d}{=} Z + V \implies a(W) \overset{d}{=} W, \tag{64}$$

where $W = Z + V$. We translate equation 64 into the condition on the pdfs as follows. For each $w \in \mathbb{R}^d$

$$f_W(w)|\det(J(w))| = f_W(w) \implies |\det(J(w))| = 1 \tag{65}$$

Substituting the above equation equation 65 into equation 62 we get

$$f_V(a(w) - a(z)) = \frac{f_V(w - z)}{|\det(J(w))|} \implies f_V(a(w) - a(z)) = f_V(w - z)$$
$$\|a(w) - a(z)\|_\alpha = \|w - z\|_\alpha, \tag{66}$$

where in the last condition in the above expression we exploit the fact that $f_V$ is a generalized Laplacian distribution. From Mazur-Ulam theorem Nica (2013) it follows that $a$ is affine. We now write $a$ as a matrix $A$ with offset vector $q$ and simplify the condition in equation 59.

For each $z \in \mathbb{R}^d$ we have

$$A(z + V) + q \overset{d}{=} Az + \hat{V} + q$$
$$\implies AV \overset{d}{=} \hat{V} \tag{67}$$
$$\implies \mathbb{E}[AVV^\mathsf{T}A^\mathsf{T}] = \mathbb{E}[\hat{V}\hat{V}^\mathsf{T}] \implies AA^\mathsf{T} = \mathsf{I}$$

Since $A$ is a square matrix and $AA^\mathsf{T}$ it follows that $A^\mathsf{T}A = \mathsf{I}$. Therefore, $A$ is an orthonormal matrix. Observe that all the elements of $\hat{V}$ are independent. Since $AV \overset{d}{=} \hat{V}$ it follows that all the elements of $AV$ are also independent. Define $AV = Q$. Observe that $A$ is an orthonormal matrix that is multiplied with a vector $V$ with all independent elements (each of which is non-Gaussian as $\alpha \neq 2$) and outputs a vector that has all independent components. From Theorem 11 in Comon (1994) we get that $A$ is a composition of permutation and scaling. Since $A$ is also orthonormal, each term in the diagonal scaling matrix can only be $1$ or $-1$. Therefore, $A = \Pi\Lambda$, where $\Pi$ is a permutation matrix and $\Lambda$ is a diagonal matrix with $+1, -1$ elements. Finally, $a(z) = \Pi\Lambda z + q$.

$\square$

### A.9 ALTERNATIVE IDENTIFICATION RESULT FOR SMALL TRANSITIONS

In this section, we analyze time series models similar to one in Hyvarinen & Morioka (2017) under the condition that the transitions are small in magnitude to arrive at permutation and scaling based identification. Let us analyze the set $\mathcal{E}^\mathsf{s}$ for this class of mechanisms. We assume that the learner knows that the mechanism is additive, and that the noise components are all independent. In the analysis below we consider bijections that are analytic (each component of the bijection is an analytic function). Consider an $a \in \mathcal{E}^\mathsf{s}$

$$a(z + V) \overset{d}{=} a(z) + \hat{V}. \tag{68}$$

We write the first-order approximation of the above identity below

$$a(z) + \mathsf{J}(z)V \overset{d}{=} a(z) + \hat{V}$$
$$\mathsf{J}(z)V \overset{d}{=} \hat{V} \tag{69}$$

where $V \overset{d}{\neq} \hat{V}$. Note that the set of solutions $a$ to equation 68 and equation 69 become equal in the limit of $\delta \to 0$, where $\delta$ is the bound on each component of $|V|$. We analyze the solution to equation 69 below.

$$\mathbb{E}[(\mathsf{J}(z)V)(\mathsf{J}(z)V)^\mathsf{T}] = \mathsf{J}(z)\mathbb{E}[VV^\mathsf{T}]\mathsf{J}(z)^\mathsf{T} = \sigma^2\mathsf{J}(z)\mathsf{J}(z)^\mathsf{T}$$
$$\sigma^2\mathbb{E}[\hat{V}\hat{V}^\mathsf{T}] = \sigma^2\mathsf{I} \tag{70}$$
$$\sigma^2\mathsf{J}(z)\mathsf{J}(z)^\mathsf{T} = \sigma^2\mathsf{I} \implies \mathsf{J}(z)\mathsf{J}(z)^\mathsf{T} = \mathsf{I} \implies \mathsf{J}(z)^\mathsf{T}\mathsf{J}(z) = \mathsf{I}$$

Since $J(z)V \overset{d}{=} \hat{V}$ and each component of $\hat{V}$ is independent, we can deduce that all the components of $J(z)V$ are independent as well. From Theorem 11 in Comon (1994), we can deduce that $J(z)$ is composed of permutation times a diagonal matrix. Since the matrix is orthonormal, each scaling component can only be $\pm 1$. We can apply this same analysis at another point $\tilde{z}$ in the neighborhood of $z$ and continue to find that the jacobian matrix is a permutation times diagonal matrix (that describes sign flips). Note that the permutation matrix times scaling used to express the Jacobian cannot change between the points $\tilde{z}$ and $z$ (if it does change then that violates the Jacobian's continuity). Since the Jacobian is equal to a fixed permutation times a fixed scaling matrix over a neighborhood, we can extend this to the entire space (here we use the fact that the $a$ is analytic and Mityagin (2015)). As a result, $a$ is of the form $\Pi \Lambda z + q$, where $\Pi$ is a permutation matrix, $\Lambda$ is a diagonal matrix.

### A.10 ANALYZING AUXILIARY INFORMATION MODELS

We now discuss how our machinery can be used in models when auxiliary information is available (Khemakhem et al. (2020a)) to arrive at permutation and scaling based identification. We define the data generation process compatible with Khemakhem et al. (2020a). Suppose the latent $Z$ is generated from a mechanism $m : \mathcal{O} \times [0,1]^d$ that takes as input some observed auxiliary information $O$ and uniform independent noise vector $U \in [0,1]^d$:

$$
\begin{aligned}
Z &\leftarrow m(O, U), \\
X &\leftarrow g(Z),
\end{aligned}
\tag{71}
$$

where $g : \mathcal{Z} \to \mathcal{X}$ is a bijection. Suppose the learner knows the mechanism. The learner selects $\tilde{g}$ and outputs $\hat{X} = \tilde{g} \circ m(o, \tilde{U})$, where $\tilde{U}$ is a random vector with each component sampled independently from $\text{Uniform}[0,1]$. The learner's goal is to match

$$
\begin{aligned}
\hat{X}|O = o &\overset{d}{=} X|O = o \\
g \circ m(o, U) &\overset{d}{=} \tilde{g} \circ m(o, \tilde{U}).
\end{aligned}
\tag{72}
$$

for all possible observations $o \in \mathcal{O}$. Define the set of solutions to equation equation 72 as $\mathcal{G}_{\text{id}}^{\text{o}}$. Consider bijection $a$ s.t. the following identity holds for all $o \in \mathcal{O}$

$$
a \circ m(o, U) \overset{d}{=} m(o, \tilde{U})
\tag{73}
$$

Define the set of all bijections $a$ that satisfy the condition in equation equation 73 as $\mathcal{E}_{\text{eq}}^{\text{o}}$. The set $\mathcal{E}_{\text{eq}}^{\text{o}}$ consists of intersection of measure preserving transformations of $Z|O = o$. Define the set of $\tilde{g}$ that are identifiable up to $\mathcal{E}_{\text{eq}}^{\text{o}}$ as $\mathcal{G}_{\text{eq}}^{\text{o}} = \{\tilde{g} \mid \tilde{g} = g \circ a^{-1}, \ a \in \mathcal{E}_{\text{eq}}^{\text{o}}\}$.

**Theorem 7.** *If the data is generated as described in equation 71, then the set of solutions to the identity in equation 72 identify true $g$ up to intersection of all the measure preserving maps in $\mathcal{E}_{\text{eq}}^{\text{o}}$ ($\tilde{g}^{-1} \sim_{\mathcal{E}_{\text{eq}}^{\text{o}}} g^{-1}$)*

*Proof.* Consider a $\tilde{g} \in \mathcal{G}_{\text{id}}^{\text{o}}$.

$$
\begin{aligned}
g \circ m(o, U) &\overset{d}{=} \tilde{g} \circ m(o, \tilde{U}) \\
\tilde{g}^{-1} \circ g \circ m(o, U) &\overset{d}{=} m(o, \tilde{u}) \\
a \circ m(o, U) &\overset{d}{=} m(o, \tilde{U})
\end{aligned}
\tag{74}
$$

From the above it follows that $\tilde{g} \in \mathcal{G}_{\text{eq}}^{\text{o}}$. Consider a $\tilde{g} \in \mathcal{G}_{\text{eq}}^{\text{o}}$.

$$
g \circ m(o, U) = g \circ a^{-1} \circ a \circ m(o, U) \overset{d}{=} \tilde{g} \circ m(o, \tilde{U})
\tag{75}
$$

From the above it follows that $\tilde{g} \in \mathcal{G}_{\text{id}}^{\text{o}}$. This completes the proof. $\qquad \square$

Let us consider additive mechanisms of the form $m(o, U) = \bar{m}(o) + U$, where $U$ has independent components. Suppose the learner knows that the mechanism is additive and the noise has independent components. The learner solves the following identity.

$$
\begin{aligned}
g \circ (\bar{m}(o) + U) &\overset{d}{=} \tilde{g} \circ (\bar{m}'(o) + \tilde{U}) \\
\tilde{g}^{-1} \circ g(\bar{m}(o) + U) &\overset{d}{=} \bar{m}'(o) + \tilde{U} \\
a(\bar{m}(o) + U) &\overset{d}{=} \bar{m}'(o) + \tilde{U}
\end{aligned}
\tag{76}
$$

Suppose that the absolute value of each component of $U$ is really small and bounded by $\delta$. We can use the first-order Taylor expansion and obtain

$$a(\bar{m}(o)) + \mathsf{J}(\bar{m}(o))U \stackrel{d}{=} \bar{m}'(o) + \tilde{U}, \tag{77}$$

where $\mathsf{J}$ is the Jacobian of $a$. Suppose the noise has zero mean. Take the expectation w.r.t. $U$ and $\tilde{U}$ on the two sides respectively to get $a(\bar{m}(o)) = \bar{m}'(o)$. Substitute this back into the equation we get that $\mathsf{J}(\bar{m}(o))U \stackrel{d}{=} \tilde{U}$. We can now follow the analysis that we carried out after equation 69 and conclude that $a$ is equal to permutation times a scaling matrix along with some offset.

## A.11 ANALYZING $\tilde{\mathcal{E}}$ WHEN $\mathcal{M} = \mathcal{M}^*$

In this section, we analyze imitators when we know the set of mechanisms that are deployed, we do not know which mechanism is used when. If $a \in \tilde{\mathcal{E}}$ and $\mathcal{M} = \mathcal{M}^*$. From the definition of $a$, it follows that for each $m \in \mathcal{M}^*$, $\exists\, m' \in \mathcal{M}^*$ such that $a \circ m = m' \circ a$. We claim that two distinct $m \in \mathcal{M}^*$ cannot share the same $m'$ (imitator). Suppose there was a common $m'$ imitating $m$ and $\tilde{m}$.

$$\begin{aligned} a \circ m &= m' \circ a \\ a \circ \tilde{m} &= m' \circ a \end{aligned} \tag{78}$$

We take the difference of the above two equations to get

$$a \circ m = a \circ \tilde{m} \tag{79}$$

Since $a$ is a bijection, we can conclude that $m = \tilde{m}$, which is a contradiction of the fact that $m$ and $\tilde{m}$ are distinct.

This claim implies that for a given $a$ there is an injective map from $\mathcal{M}^*$ to $\mathcal{M}^*$. If the set $\mathcal{M}^*$ is finite, then from Pigeonhole principle it follows that this injective map is a bijection.

Let us index the mechanism $\mathcal{M}^* = \{m^1, \cdots, m^n\}$. We call the bijection map $\pi : \mathcal{M}^* \to \mathcal{M}^*$

Consider the element $i$. We claim $\exists\, l \in \{1, \cdots, n\}\ \pi^l(i) = i$. We write the chain starting from $i$ as $i \to \pi(i) \to \pi^2(i), \cdots, \pi^k(i)$. Since the chain $(\pi(i) \to \pi^2(i), \cdots, \pi^k(i))$ has $n$ steps there have to be at least two elements that are equal. Suppose $p > q$ and $\pi^p(i) = \pi^q(i)$

$$\pi^p(i) = \pi^q(i) \implies \pi^{p-1}(i) = \pi^{q-1}(i) \implies \cdots ..\pi^{p-q}(i) = i \tag{80}$$

In the above at each step we use the fact that $\pi$ is a bijection and that shows the claim that $\pi^l(i) = i$. We now use this observation to carry out the following simplification

$$\begin{aligned} m^i &= a^{-1} \circ m^{\pi(i)} \circ a \\ m^{\pi(i)} &= a^{-1} \circ m^{\pi^2(i)} \circ a \\ &\vdots \\ m^{\pi^k(i)} &= a^{-1} \circ m^i \circ a \end{aligned} \tag{81}$$

Substituting the second equation $m^{\pi(i)}$ into first, and then the third $m^{\pi^2(i)}$ and so on we get

$$m^i = a^{-k} \circ m^i \circ a^k \tag{82}$$

Therefore, for each $m^i$, $\exists\, k$ such that $a^k$ is its equivariance.

To summarize, if $a \in \tilde{\mathcal{E}}$ and $\mathcal{M} = \mathcal{M}^*$, where $\mathcal{M}$ is a finite set, then for each mechanism $m \in \mathcal{M}$, $\exists\, k \in \{1, \cdots, |\mathcal{M}|\}$ such that $a^k$ is its equivariance.

## A.12 TRANSLATING IDENTITY INTO LOSS FUNCTIONS AND PRELIMINARY EXPERIMENTS

We restate the two loss choices based on our identity $\tilde{g} \circ m \circ \tilde{g}^{-1}(x_t) = x_{t+1}$ below.

- **Loss based on observations.** The identity above immediately implies an autoencoder-style algorithm where one minimizes a reconstruction loss of the form

$$\min_{\tilde{g} \in \mathcal{H}, \tilde{h} \in \mathcal{H}, \tilde{m} \in \mathcal{M}} \sum_t \mathbb{E}\Big[ \|X_{t+1} - \tilde{g} \circ \tilde{m} \circ \tilde{h}(X_t)\|^2 \Big] \qquad (83)$$

where $\mathcal{H}$ is the hypothesis class of functions for $\tilde{g}$ and $\mathcal{M}$ is the hypothesis class of mechanisms.

- **Loss based on latents.** Alternatively, one could re-write the observation identity as $g^{-1} \circ x_{t+1} = m \circ g^{-1} \circ x_t$ and use a contrastive loss as follows

$$\min_{\tilde{g}} \sum_t \mathbb{E}\Big[ -\log \Big( \frac{\tilde{g}(X_{t+1})^{\mathsf{T}} \tilde{m} \tilde{g}(X_t)}{\tilde{g}(X_{t+1})^{\mathsf{T}} \tilde{m} \tilde{g}(X_t) + \sum_\tau \tilde{g}(X_\tau)^{\mathsf{T}} \tilde{m} \tilde{g}(X_t)} \Big) \Big] \qquad (84)$$

where $\tau$ represents other time instances, i.e., $\tau \neq t + 1$. The positive pair in the contrastive loss is formed by the adjacent time instances and the negative pair is formed by non-adjacent time instances. Identifying which of these two losses works better in practice is an important empirical question. Note how in both the losses above, we have explicitly not enforced invertibility for the learned g.

We present our initial experiments on 3dIdent dataset from [Zimmerman et al. 2021], using the contrastive loss described above. With contrastive pairs generated by a (fixed) random orthogonal matrix U applied to the latents, we obtain the following values for linear disentanglement score ($R^2$ of the predictions of the true representation using a linear model). We report median scores over 10 seeds.

- **Standard contrastive learning.** Linear disentanglement score of 0.29

- **Contrastive leveraging with exact mechanism knowledge.** Leveraging the true U as the mechanism in the contrastive loss achieves a median score of 0.76.

- **Contrastive leveraging with some knowledge of mechanism** If we use a random orthogonal matrix $\tilde{U} \neq U$, that achieves a median score of 0.64. The random matrix can be interpreted as sampling a random $m$ from the hypothesis class $\mathcal{M}$; it seems likely that better results are possible by optimizing over $\mathcal{M}$.

