# OpenReview forum: "Properties from mechanisms: an equivariance perspective on identifiable representation learning"
_ICLR.cc/2022/Conference — ICLR 2022 Spotlight_

### Official Review · Reviewer_bAsz · 2021-11-01

**Correctness:** 4
**Technical Novelty And Significance:** 3
**Empirical Novelty And Significance:** Not applicable
**Recommendation:** 6
**Confidence:** 3

**Main Review:**

# Strengths

+ This idea that an identified low-level mechanism can be used to disentangle high-dimensional observations is to my knowledge novel. It seems quite applicable particularly in control from pixel inputs.
+ I checked the first couple of proofs in the appendix to get a feel for the techniques used. They looked correct.
+ The main figure is helpful and very clear.
+ The examples that are given, for example, the linear mechanism setting and the equivariances of affine mechanisms, are very helpful for understanding.
+ Using the identifiability results to give a result from previous work was a nice illustration of the generality of the setup.

# Weaknesses

- No algorithm is proposed that actually uses the identifiability results, even in the simplest setting where the mechanism is totally known. I understand that this is not the focus of this paper, though it would have been nice to demonstrate the utility of these results and show that the assumptions are reasonable.
- I think the bijectivity assumption is not unreasonable here but could have been better justified. One might argue that the assumption does not allow us to consider, for example, acceleration in the latent state (since a change in acceleration is not reflected in a single image). Injectivity is also assumed in the block MDP literature but is justified by the (common practice in RL) of frame-stacking, which would result in injectivity holding for variables like acceleration. The proofs I looked at leaned heavily into the bijectivity assumption.
- I think a big part of the interest in disentanglement comes from not only learning the representation but also how the latent variables evolve.   This aspect is lost here since we either assume a known mechanism or assume a class of possible mechanisms but do not try to identify the mechanism. Nevertheless, I think the approach is still valuable because it 1) seems quite orthogonal to other works which try to make use of, say, conditioning on side information 2) seems to be potentially practical in simple control settings with high-dimensional observations.
- The bijectivity and known mechanism assumption means some of the early theorems are not too surprising. This does not mean that they are not valuable, however.
- It was not clear to me if similar results could be obtained if the observations are observed with additive noise.

# Questions
- Can more related works beyond the Klindt 2020 paper be explained within this framework? If not, why?

# Minor points:
- "the success of pre-trained of" typo.
- Figure 1: can be made clearer on the diagram that the red latent representation does not give a correct prediction of the next observations.
- "we also have have to"  typo
- A1 contains no text. It should reference the associated table.



**Summary Of The Paper:**

This work gives several identifiability results for representation learning from sequences of observed variables when the mechanism governing the underlying latent variables is known, and the observation renderer is bijective. The identifiability of an inverse renderer is shown to depend only upon the equivariances of the mechanism(s) governing the evolution of the underlying latent variables. Extensions are given to the case of a known mechanism class and to stochastic mechanisms.



**Summary Of The Review:**

I think this is an exciting direction of work that runs orthogonally to a lot of the recent advances in disentanglement, by assuming knowledge of the underlying mechanism rather than independence and distribution assumptions on the latent variables. Presentation of an algorithm or some basic experiments to justify the feasibility of the new setting would substantially strengthen the paper. However, I think the identifiability results are useful enough for follow-up work that they have merit on their own.

---

### Official Review · Reviewer_7um5 · 2021-11-02

**Correctness:** 3
**Technical Novelty And Significance:** 3
**Empirical Novelty And Significance:** Not applicable
**Recommendation:** 8
**Confidence:** 4

**Main Review:**

**Strengths.**
- By approaching the problem through the lens of equivariances of the underlying mechanism(s) relating subsequent observations, the paper provides an interesting perspective on an important topic (identifiable representation learning) that (as far as I can tell) is novel.
- The emphasis on non-iid observations related through some shared latent dynamics shares a common ground with many of the recent results in the literature (discussed in section 5) which makes it conceptually appealing. The lack of independence assumptions is also refreshing and makes this work of potential interest for causal representation learning.
- The theoretical results seem sound (though I only checked the proofs of Thm.1 and Corollary 1 in detail)
- The paper is well written and, despite being quite technical in nature, accessible and not too hard to follow; the organisational structure and succession of results make sense. I particularly liked the motivating bouncing balls example which is picked up repeatedly, especially in the first half of the paper, to illustrate assumptions or conditions (this was very helpful and could perhaps feature more prominently in the latter part of the paper).
- Related work and limitations are adequately discussed in sections 5 and 6.

**Weaknesses.**
(some of these are also acknowledged by the authors, see section 6)
- The paper assumes the reconstruction task to be solved and assumes invertibility not only for true $g$ but also for the learnt model; both of these are limitations in practice, and it would be nice to discuss how some insights from the paper may translate into practical learning rules.
- The paper does not present any empirical results or experimental analysis. This is partly justified by being clearly a theory paper, but some aspects could also be probed empirically: e.g., by rendering some data using one of multiple known mechanisms, training a (Slow)VAE-type model that has access to this knowledge about mechanisms, and checking how the learnt decoder compares to the true one.
- While it is pointed out that not knowing the exact mechanism and using a larger hypothesis class can be a source of non identifiability, it would be nice to quantify this somehow, either theoretically (e.g., as a function of the capacity of the function class) or empirically.
- Some statements (e.g., Theorem 1) are a bit hard to parse, and others seem to be slightly imprecise (e.g., the definition of $\tilde{\mathcal{G}}_{id}$ after eq. (6): isn't this missing the qualifiers $\forall t, \exists \tilde{m}_t$? and the set $\mathcal{E}$ defined on p.4 depends on $m$ which should probably be highlighted notationally)

**Other comments / suggestions.**
- Regarding the section on stochastic mechanisms, I was wondering whether there could be a connection to measure-preserving maps / automorphisms which are a common source of nonidentifiability in nonlinear ICA?
- Since the paper clearly tries to make an effort on discussing connections to existing results, it may be worth also comparing with and discussing the following works which seem relevant as they also revolve around deriving identifiability results for learning from multi-view data:
- - [Gre19] The Incomplete Rosetta Stone Problem: Identifiability Results for Multi-View Nonlinear ICA. Luigi Gresele, Paul K. Rubenstein, Arash Mehrjou, Francesco Locatello, Bernhard Schölkopf. https://arxiv.org/abs/1905.06642
- - [Von21] Self-Supervised Learning with Data Augmentations Provably Isolates Content from Style. Julius von Kügelgen, Yash Sharma, Luigi Gresele, Wieland Brendel, Bernhard Schölkopf, Michel Besserve, Francesco Locatello. https://arxiv.org/abs/2106.04619
- The notation for composition is not consistently used: sometimes $a\circ g(x)$ is used, sometimes $a\circ g\circ x$

**Typos.**
- p.3, 1st para: "have have"
- p.4 1st para: "any an encoder"
- p.4 after eq. (3): "the follow functions"
- p.4 after eq. (4): "that that"
- p.5 sec 2.2.1, 1st sentence: missing a pronoun like "it"
- p.7 1st para: "it's" --> "its"; same sentence: "diverse SET of"; next sentence: "in THE bouncing"
- p.8 1st sentence: shouldn't this refer to eq. (8) instead of (7)?
- p.8, after Thm. 4: "consistent WITH the"
- p.12, A.2: eq. 12 misses two equality signs at the start of the second and third line; the reference after eq. (12) I believe should be to equation (4), not (2)




**Summary Of The Paper:**

**Scope.** The paper studies unsupervised/weakly-supervised representation learning from an identifiability perspective. The considered problem setting is that observations $x_t$ are generated from some bijective function $g$ applied to latent codes $z_t$, $x_t=g(z_t)$; and that we have access to successive observations $..., x_t, x_{t+1}, ...$ which are related through shared *mechanisms* $m_t$ dictating the dynamics in the latent space, $z_{t+1}=m_t(z_t)$. The paper then provides a (purely) theoretical analysis regarding the identifiability of $g$ given (partial) knowledge of the mechanisms $m_t$. In particular, the identifiability class of $g$ is derived in terms of *equivariances* of the mechanisms.

**Results.** For the case of a single known mechanism $m$, it is shown that $g$ can be recovered only up to equivariances of $m$, i.e., up to bijections $a$ that commute with $m$, meaning that any learnt function $g'$ relates to the true $g$ via $g'=g\circ a$ for some $a$. This result is strengthened when multiple distinct known mechanisms govern the dynamics, where identifiability is obtained up to shared equivariances, i.e., such $a$ that commute with all $m_t$. When the relevant mechanisms are not known, non-identifiability issues increase with the size of the hypothesis class which is attributed to the presence of so-called *imitators*, transformations that give rise to the same behaviour as some mechanism when combined with another mechanism. Finally, a result is presented for the case of stochastic mechanisms which give rise to conditional transition distributions $p(z_{t+1}|z_t)$: here, equivariance and imitation are relaxed to hold only in distribution. The insights are used to re-derive a known identifiability result by Klindt et al. (2020).

**Summary Of The Review:**

Overall, this is an interesting paper and a valuable contribution to ICLR. I think that the lack of empirical results and experiments is justified by the theoretical contribution. Subject to some of the further improvements in aspects of the presentation, as well as discussion of additional related work suggested in my main review, I recommend acceptance.

---

### Official Review · Reviewer_muVw · 2021-11-05

**Correctness:** 4
**Technical Novelty And Significance:** 4
**Empirical Novelty And Significance:** Not applicable
**Recommendation:** 6
**Confidence:** 2

**Main Review:**

This is an insightful theory paper. Rather than making arbitrary independence assumptions on the latents, it considers the true data-generating mechanism and attempts to recover the latent based on assumptions on knowledge of the mechanism. While some of the results are intuitively mentioned in prior empirical work and may be obvious (i.e., the latents are unidentifiable upto scaling constant), I think the paper extends it to provide a strong result when there are multiple mechanisms at play: in that case, identification may become easier (cor 1). The constraints are shared and should be satisfied across all mechanisms.

To connect with realistic situations, I liked the extension to the unknown mechanism and stochastic mechanisms. Although I have to say that I did not check the appendix for proof, so I would defer to other reviewers on the theoretical validity.

My main feedback is on the relevance of this formulation to representation learning tasks:
1. Besides the balls example, can the authors provide ML tasks where the time-based mechanism would make sense? I presume it is not useful for images, but applicable to videos. What about natural language? Is there a similar mechanism? I'm trying to understand the empirical problems where mechanism-based characterization can help.
2. I would have liked to see a simple implementation of the idea (even a toy one would suffice). In the balls example, would following the proposed method recover the Newton's laws coefficients, given some representation of the image/state?  I'm guessing you will need to assume a reconstruction loss, but for a simple problem, that choice should not matter.
3. Finally, the assumption of invertible function is a limitation, as the authors note.




**Summary Of The Paper:**

The paper uses the knowledge of generating mechanisms to identify latent representation for given data. It contributes to the question of identifiability in unsupervised learning. It relaxes the knowledge of generating mechanism to unknown mechanism, and also from deterministic mechanism to probabilistic mechanism.

**Summary Of The Review:**

Good theoretical insights, can have better connection to practice

---

### Decision · Program_Chairs · 2022-01-20

**Decision:**

Accept (Spotlight)

**Comment:**

The paper provides new insights about how to identify latent variable distributions, making explicit assumptions about invariances. A lot of this is studied in the literature of non-linear ICA, although the emphasis here is on dropping the "I". I think more could be said about how allowing for dependencies among latents truly change the nature of the problem since any distribution can be built out of independent latents, by some more explicit contrast against the recent references given by the reviewers. In any case, the role of allowing for dependencies in the context of the invariances adopted is discussed, and despite no experimentation, the theoretical results are of general interest to the ICLR community and a worthwhile contribution to be discussed among researchers in this field.